behaviour, evolution, theoretical biology

inequality, reproductive skew, mating systems, social evolution, phylogenetic comparative analysis

**Author for correspondence:**
Cody T. Ross
e-mail: cody_ross@eva.mpg.de

# The multinomial index: a robust measure of reproductive skew

Cody T. Ross[1], Adrian V. Jaeggi[2], Monique Borgerhoff Mulder[1,3], Jennifer E. Smith[4], Eric Alden Smith[5], Sergey Gavrilets[6] and Paul L. Hooper[7]

[1]Department of Human Behavior, Ecology and Culture, Max Planck Institute for Evolutionary Anthropology, Leipzig, Germany
[2]Institute of Evolutionary Medicine, University of Zurich, Zurich, Switzerland
[3]Department of Anthropology, University of California, Davis, CA, USA
[4]Department of Biology, Mills College, Oakland, CA, USA
[5]Department of Anthropology, University of Washington, Seattle, WA, USA
[6]Departments of Mathematics and Ecology & Evolutionary Biology, Center for the Dynamics of Social Complexity, and National Institute for Mathematical and Biological Synthesis, University of Tennessee, Knoxville, TN, USA
[7]Santa Fe Institute, Santa Fe, NM, USA

  CTR, 0000-0002-0067-4799; AVJ, 0000-0003-1695-0388; MBM, 0000-0003-1117-5984; JES, 0000-0002-3342-4454; EAS, 0000-0002-9482-9666; SG, 0000-0003-1581-4018; PLH, 0000-0002-4673-7513

Inequality or skew in reproductive success (RS) is common across many animal species and is of long-standing interest to the study of social evolution. However, the measurement of inequality in RS in natural populations has been challenging because existing quantitative measures are highly sensitive to variation in group/sample size, mean RS, and age-structure. This makes comparisons across multiple groups and/or species vulnerable to statistical artefacts and hinders empirical and theoretical progress. Here, we present a new measure of reproductive skew, the multinomial index, $M$, that is unaffected by many of the structural biases affecting existing indices. $M$ is analytically related to Nonacs' binomial index, $B$, and comparably accounts for heterogeneity in age across individuals; in addition, $M$ allows for the possibility of diminishing or even highly nonlinear RS returns to age. Unlike $B$, however, $M$ is not biased by differences in sample/group size. To demonstrate the value of our index for cross-population comparisons, we conduct a reanalysis of male reproductive skew in 31 primate species. We show that a previously reported negative effect of group size on mating skew was an artefact of structural biases in existing skew measures, which inevitably decline with group size; this bias disappears when using $M$. Applying phylogenetically controlled, mixed-effects models to the same dataset, we identify key similarities and differences in the inferred within- and between-species predictors of reproductive skew across metrics. Finally, we provide an R package, SkewCalc, to estimate $M$ from empirical data.

## 1. Introduction

The unequal distribution of reproduction within a group—a feature also known as reproductive skew—is common across many animal societies [1]. In some cases, a small same-sex fraction of a population obtains the majority of reproductive output (high reproductive skew), whereas in other cases reproduction is more equally distributed across same-sex individuals (low reproductive skew) [2]. Skewed patterns of reproduction emerge in taxa as diverse as social insects [3], rodents [4], birds [5], social carnivores [6,7], non-human primates [8], and humans [9].

The intensity of natural and sexual selection is tightly linked to levels of variation in mating access and offspring production [10–12]. As such, effective measurement of reproductive skew/inequality lies at the heart of building an

empirical understanding of the dynamics of natural and sexual selection [13–15]. Identification of the factors favouring the emergence of reproductive inequality has also been a key priority for understanding the evolution of social and mating systems, as well as the dynamics of sex differences in parental care and competitive traits [16]. The link between reproductive skew and social behaviour, in particular, has attracted much theoretical attention, motivating the derivation of models aimed at explaining variation in reproductive levelling *within* groups [2,13,17–21]. Moreover, measures of inequality in reproduction, wealth, status, and other fitness-relevant characters are emerging as a core focus of comparisons *between* human and non-human social systems [22–26], leading to more generalized explanatory models for the structuring of inequality across animal societies, both human and non-human [15]. In humans specifically, we are starting to theorize and test models that dynamically link ecological context and resource defensibility [27], marriage and inheritance systems [28–30], and reproductive skew [31,32].

Although the topic of reproductive inequality has been of continuing theoretical importance to evolutionary biologists and social scientists, the current toolkit available for quantifying such inequality proves insufficient in practice, and measurement problems have therefore generated much debate [33–37]. Though the details of a given research question may sometimes necessitate other kinds of measures, researchers across fields as diverse as biology, anthropology, and economics generally agree on the following desiderata for comparative measures of skew/inequality: they should be unitless [38], robust to variation in sample/group size between study populations [39], control for the effects of heterogeneity in age or 'exposure time' to risk of reproduction [35], and be related to standard measures of variance in reproductive success (RS) [40]. Studies attempting to compare the strength of natural selection across species [26], determine sex differences in the intensity of sexual selection [15,41], identify variation in the extent of reproductive skew across groups/species [42–44], or uncover associations between covariates (such as group size) and reproductive skew [8,45] can come to spurious conclusions if the inequality/skew measure being used does not simultaneously meet these criteria.

Existing measures of reproductive skew generally trade-off one desideratum for another. For example, the opportunity for selection index [40], $I$, and its sampling adjusted counterpart Morisita's index [46,47], $I_\sigma$, are unitless, invariant to sample size, and related analytically to variance in RS, but are sensitive to age-structure; $I$ is even sensitive to mean RS [48]. In contrast, Nonacs' binomial index [35], $B$, accounts for age-structure, but introduces a strong statistical bias based on sample/group size that has gone largely unaddressed in the literature. This issue is particularly problematic in cross-species (or crosscultural) comparisons where group size and/or sample size can vary *substantially*. In short, despite the centrality of reproductive inequality in a range of models across the evolutionary sciences, there is still no reliable measure of reproductive skew that permits rigorous comparative research to evaluate the predictions of such models.

In this paper, we derive a new metric of skew/inequality in reproductive rate, and a Bayesian method of quantifying uncertainty in this measure, from a simple set of first principles. We then demonstrate that this metric meets the desiderata described above. The rest of the paper runs as follows: in §2, we outline the theoretical importance of comparative studies of reproductive inequality/skew to central questions across evolutionary biology, economics, and anthropology. In §3, we introduce the multinomial index, $M$. In §4, we provide a detailed mathematical derivation of $M$ and then compare $M$ to several existing skew measures through the analysis of 270 000 simulated RS datasets with differing input parameters for age-structure, group size, mean RS, and skew. This analysis shows that $M$ is unaffected by the statistical biases that have affected other measures of skew. We also introduce an R package for calculating Bayesian posterior estimates of $M$, and illustrate how skew estimates can be compared between populations while accounting for posterior uncertainty due to differences in sample size and RS rate. Specifically, we draw on census data from three small-scale human populations with different marriage systems and population sizes to show how Bayesian estimates of $M$ appropriately disentangle effect size and posterior credibility. In §5, we detail the analytic relationship between $M$ and other measures of skew, so that researchers can draw on a large body of published skew estimates and convert them into a standard and comparable form. In §6, we conduct an illustrative phylogenetically controlled, multi-level comparative analysis of skew in male primates using previously published data. In §7, we conclude by discussing the broad usefulness of the multinomial index for future comparative research on reproductive skew and other forms of inequality.

## 2. Skew in a comparative context

Biological populations can differ greatly in the level of inequality characterizing the distribution of reproduction across samesexed individuals [8]. In humans, reproductive inequality often varies substantially among cultural groups [9], especially as a function of marriage system and material wealth inequality. This topic has been of keen interest to evolutionary minded economists and anthropologists [28,29,49,50], who argue that the coevolutionary rise of monogamy, reproductive levelling, and highly unequal agrarian-state social structures constitutes one of the most striking counter-examples to otherwise wellaccepted fitness/utility-based models of reproductive decision-making, like the polygyny threshold model [51]. Resolution of this paradoxical empirical pattern may be explained by norms for reproductive levelling [52–55] that enhance food security, group functionality, and/or success in intergroup competition [56–58], norms for monogamous partnering [29,50,59–61], or the level of complementarity in returns to biparental investment in humans [61,62]. Tests of such predictions, however, require comparative datasets and unbiased skew measures.

Beyond humans, Johnstone [2] and Kutsukake & Nunn [8] argue that a large body of theory on reproductive skew predicts clear relationships between inequality in reproduction and various social, ecological, and genetic factors—including relatedness, ecological constraints on reproduction, and opportunities to suppress or control the reproductive activities of other individuals. Differences in reproductive skew are thus predicted to have wide-reaching consequences for the evolution of biological characteristics (e.g. ornamentation [63], and testes size [64]), as well as social and behavioural ones (e.g. stable group size [65], effective population size [48], male tenure length [1], sociality [66], and the patterning of violence [67] and aggression [68]). To effectively test such theory, however,

cross-species or cross-genera comparisons are often needed, but they have also been relatively sparse (but see [1,8]).

In one of the widest-scale comparative studies of reproductive skew to date, Kutsukake & Nunn [8] investigate the cross-species patterning of reproductive skew in male primates as a function of a suite of covariates. The data here are strong: sex-specific reproductive behaviour has been well-studied across primate species, and primates possess the requisite variation in social systems, mating systems, and ecological setting needed to compare competing predictions [69]. However, even within a small clade like primates, estimating differences in reproductive skew across species introduces some unique challenges: differences in age-structure, group size, and mean reproductive rate can preclude statistical comparisons based on existing skew metrics. In §6, we show how biased skew metrics can confound inference in this comparative study and others like it. To remedy these issues, we introduce a new metric of reproductive skew—the multinomial index, $M$—that will facilitate wider-scale comparative research.

## 3. A comparable measure of skew

### (a) Definition

Assuming we have data on an RS measure, $r$, and some age or 'exposure time' measure, $t$, from a sample of $N$ individuals, then $M(r, t)$ is defined as:

$$M(r, t) = \check{M}(r, t) - \mathbb{E}[\check{M}(X, t)], \tag{3.1}$$

where:

$$\check{M}(r, t) = \frac{N}{R^2} \sum_{i=1}^{N} (r_i - \bar{r}_i)^2 \tag{3.2}$$

and where:

$$X \sim \text{Multinomial}\left(R, \frac{t}{T}\right). \tag{3.3}$$

Equation (3.1) defines $M(r, t)$ to be the difference of the observed estimate of $\check{M}(r, t)$ from its expected value if RS were distributed as a multinomial outcome with the same sample size, average RS rate, and exposure time vector. Equation (3.2) then defines $\check{M}(r, t)$, an extension of the opportunity for selection [10,40], $I$, that adjusts for unequal exposure time to risk of RS. $R$ is the total number of offspring produced by a sample of $N$ individuals, $T$ is the total exposure time contributed by all $N$ individuals, $r_i$ is the number of offspring produced by individual $i$, and $\bar{r}_i = (R/T)t_i$ is the expected number of offspring that individual $i$ would have produced at his or her age if reproductive rates were perfectly equal within the group. Interpretation of $M$ is similar to that of $B$: $M = 0$ means that RS is distributed as expected under a random multinomial model with equal RS rates, $M > 0$ means that reproduction is positively skewed and $M < 0$ means that reproduction is shared more equally than expected under a random multinomial model with equal RS rates.

Like $B$ [35,36], $M$ accounts for variation in the amount of time that individuals are at risk of reproducing and it remains analytically related to other common measures of skew/inequality (see §5). The analytic relationship between $M$ and these other measures should advance cross-population analyses of reproductive skew by allowing researchers to draw on a large published literature of skew values and compare them within a standard framework. Advantageously—and in contrast to $B$—$M$ is not biased by differences in sample or group size. This is important because, as we show later, variation in group size has confounded past efforts to compare reproductive skew across populations.

### (b) Qualifications and interpretation

Despite its comparative robustness relative to existing measures, $M$, is not universally applicable to all questions about reproductive skew; specific skew indices should be carefully chosen with respect to the scientific questions being addressed [34,48]. Moreover—as is also true when using other metrics—researchers using $M$ must select sampling frames, RS proxies, and exposure time measures that are relevant to their research questions. When applied to observational data on fertility and age, $M$ specifically measures heterogeneity in fertility rates among individuals. If heterogeneity in fertility rates among living individuals is not the target of inference, then inputs to $M$ can be changed to better address the research question. For example, by defining the sampling frame to include only individuals of a given cohort with completed reproductive histories—i.e. those born in a given year and who are now either deceased or post-reproductive—and then fixing exposure time to a constant, $M$ will reflect heterogeneity in fertility per unit lifetime, rather than per unit year.

More generally, care should be taken regarding both sampling frame and function inputs. RS data can be defined as offspring ever produced—reflecting inequality in fertility—or as offspring recruited to reproductive age—reflecting inequality in both fertility and recruitment. Data on age/exposure time may be passed into $M$—so that $M$ reflects inequality in reproductive rate while living—or age/exposure time may be held fixed—so that $M$ reflects inequality in lifetime RS. The subsets of data fed into $M$ must also be considered—e.g. should the data be limited to individuals from the same birth cohort, or all individuals alive and of reproductive age at a given point in time? If a complete census of individuals is not constructed, estimates of skew might be impacted by sampling design and/or dropout due to differential mortality (see [48] for a review of possible issues). In short, estimates of $M$ will reflect different quantities based on the choice of input variables, sampling design, and other data inclusion criteria, as is necessarily true of any existing or potential skew measure.

Inputs to $M$ may also be purposefully modulated, and the resultant $M$ values compared, to learn more about the components of skew. If, for example, fertility and adult mortality are positively correlated due to trade-offs in growth versus reproduction, exposure-time-adjusted $M$ would be higher than $M$ calculated with exposure time held fixed; exposure-time-adjusted $M$ would detect that high-mortality, high-fertility phenotypes reproduce at a higher rate compared to low-mortality, low-fertility phenotypes. Measuring $M$ in both ways would unpack how the components of reproductive inequality—i.e. differential fertility, versus differential survival—vary across groups.

## 4. Derivation

Let $R$ be the total number of offspring produced by a sample of $N$ individuals, and $r_i$ be the number of offspring produced

by individual $i$, then $R = \sum_{i=1}^{N} r_i$. The share (or fraction) of the total number of offspring produced by individual $i$ is then $\hat{r}_i = r_i / R$. Next, let $t_i$ be the amount of time that individual $i$ spends in the group (we will refer to this as *exposure time to risk of RS*). The sum of the exposure time of all group members is then $T = \sum_{i=1}^{N} t_i$, and the share (or fraction) of exposure time attributable to individual $i$ is $\hat{t}_i = t_i / T$. If all individuals are present for an equal amount of time, then $\hat{t}_i = 1/N$, for example.

If the expected reproductive rate (or the probability of producing offspring per unit time) were equal across individuals in the group, the expected number of offspring produced by individual $i$ would be equal to $\bar{r}_i = R\hat{t}_i = (R/T)t_i$. The following expression thus measures the average squared deviation of observed reproductive success from that expected if reproductive rates were equal:

$$\frac{1}{N} \sum_{i=1}^{N} (r_i - \bar{r}_i)^2. \tag{4.1}$$

Equation (4.1), however, depends on the unit of measurement (i.e. squared mean reproductive success). This problem can be fixed in a standard way (e.g. as with the $I$ index) by normalizing by the square of mean RS, $R^2/N^2$. This results in the 'raw' or 'uncorrected' multinomial index, $\check{M}$, which is the normalized average squared deviation from proportional reproductive success. This metric can be expressed equivalently in many forms, including as a standardized conditional variance, as we show in the step-by-step derivation in electronic supplementary material, section 1:

$$\check{M}(r, t) = \frac{N^2}{R^2} \frac{1}{N} \sum_{i=1}^{N} (r_i - \bar{r}_i)^2 \tag{4.2a}$$

$$= N \sum_{i=1}^{N} (\hat{r}_i - \hat{t}_i)^2 \tag{4.2b}$$

$$= \frac{N^2}{R^2} [\mathrm{var}(r) - \mathrm{var}(R\hat{t})] \tag{4.2c}$$

$$= \frac{N^2}{R^2} \mathrm{var}(r)[1 - \mathrm{corr}(r, R\hat{t})^2] \tag{4.2d}$$

$$= \frac{N^2}{R^2} \mathbb{E}[\mathrm{var}(r|\hat{t})]. \tag{4.2e}$$

The standard normalization, however, leaves a negative dependency between mean RS and $\check{M}$ (as is seen also with $I$ [48]), due to the fact that any sample estimate of reproductive inequality must increase as $R$ decreases below $N$, even if reproduction in the true generative model were described by equal rate parameters. Refer to figure 1 for a visualization of this behaviour.

This dependency can lead to measurement problems, since $\check{M}$ values will differ due to sampling variation, especially in small samples and in samples with low rates of RS (see similar discussion in [48]). To generate a measure of skew unbiased by mean RS, we first specify the form of the bias function and then measure the expected offset of $\check{M}(r, t)$ from the bias to yield our 'corrected' metric, $M$. Let $X$ have a multinomial distribution with size parameter $R = \sum_{i=1}^{N} X_i$ and probability vector $\hat{t}$. Then $\mathbb{E}[\check{M}(X, t)]$ gives the bias—the expected level of normalized variance in reproductive rate observed when the underlying rate of reproduction across individuals is, by fiat, actually equal.

To remove the bias introduced by sampling, we define:

$$\begin{aligned} M(r, t) &= \mathbb{E}[\check{M}(r, t) - \check{M}(X, t)] \\ &= \check{M}(r, t) - \mathbb{E}[\check{M}(X, t)]. \end{aligned} \tag{4.3}$$

$M(r, t)$ measures the extent to which observed reproductive skew differs from the level of skew expected under a generative multinomial model with equal reproductive rates. The absolute measure of skew might be more relevant for some problems—for example, if change in reproductive skew is measured using full census data from a single population over time, see [48]—so use of $\check{M}(r, t)$ or $M(r, t)$ needs to be based on context.

## (a) The nonlinear effects of age

The basic derivation of $M$ assumes that the risk of reproduction is independent of age—or equivalently, that all years in the life course contribute equally to risk of reproductive success. In some populations, especially human populations, however, this assumption is likely to be violated. If we assume that the relationship between age and expected reproductive success over a time interval can be measured using an elasticity parameter, $\beta \in (0, 1)$, where: $\mathbb{E}[r_i|a_i, b_i] = \alpha(b_i^\beta - a_i^\beta)$. Then we can write a more general definition for $\check{M}(r, t)$ as $\check{M}(r, a, b, \beta)$, where $a_i$ is the age of individual $i$ at first observation and $b_i$ the age at death or censor. This is:

$$\check{M}(r, a, b, \beta) = N \sum_{i=1}^{N} \left( \hat{r}_i - \frac{b_i^\beta - a_i^\beta}{\sum_{j=1}^{N} (b_j^\beta - a_j^\beta)} \right)^2. \tag{4.4}$$

Empirically, the parameters of the model—$\alpha$ and $\beta$—can be estimated on a population-by-population basis using the same data needed to estimate $M$ itself. In some cases, the conditional expected value of $r_i$ given $a_i$ and $b_i$ may be a highly nonlinear function of age, not well-modelled by either a simple proportionality assumption or by a diminishing marginal returns assumption. To generalize the effect of age to arbitrary functional forms, we present a more robust Gaussian Process estimation procedure in the electronic supplementary material, section 2, and include this model in the `SkewCalc` package.

## (b) Bayesian inference

Both $\check{M}$ and $M$ as expressed above are point estimates that do not reflect the uncertainty inherent in their calculations. When $N$ is small, for example, estimates are less credible; downstream analysis of skew indices should account for such uncertainty. To estimate the posterior probability distributions of $\check{M}$ and $M$ given a specific dataset, we provide a Bayesian estimation procedure in the electronic supplementary material, section 3. The `SkewCalc` package employs this procedure in `R` and estimates the relevant posterior distributions from individual-level data on RS and exposure time using a simple user interface. Vignettes are provided in the `SkewCalc` package. Further software details are provided in the electronic supplementary material, section 4.

Here, we illustrate both methods of estimating $M$—with point estimates and through Bayesian inference—using data taken from three human populations—polygynous Kipsigis ($n = 848$ males and 1239 females) [70], serially monogamous Afrocolombians ($n = 91$ males and 142 females), and monogamous Emberá ($n = 25$ males and 30 females) [71]. Figure 2(a) shows that $M$ is structured as would be expected given the marriage system—with male $M$ values decreasing

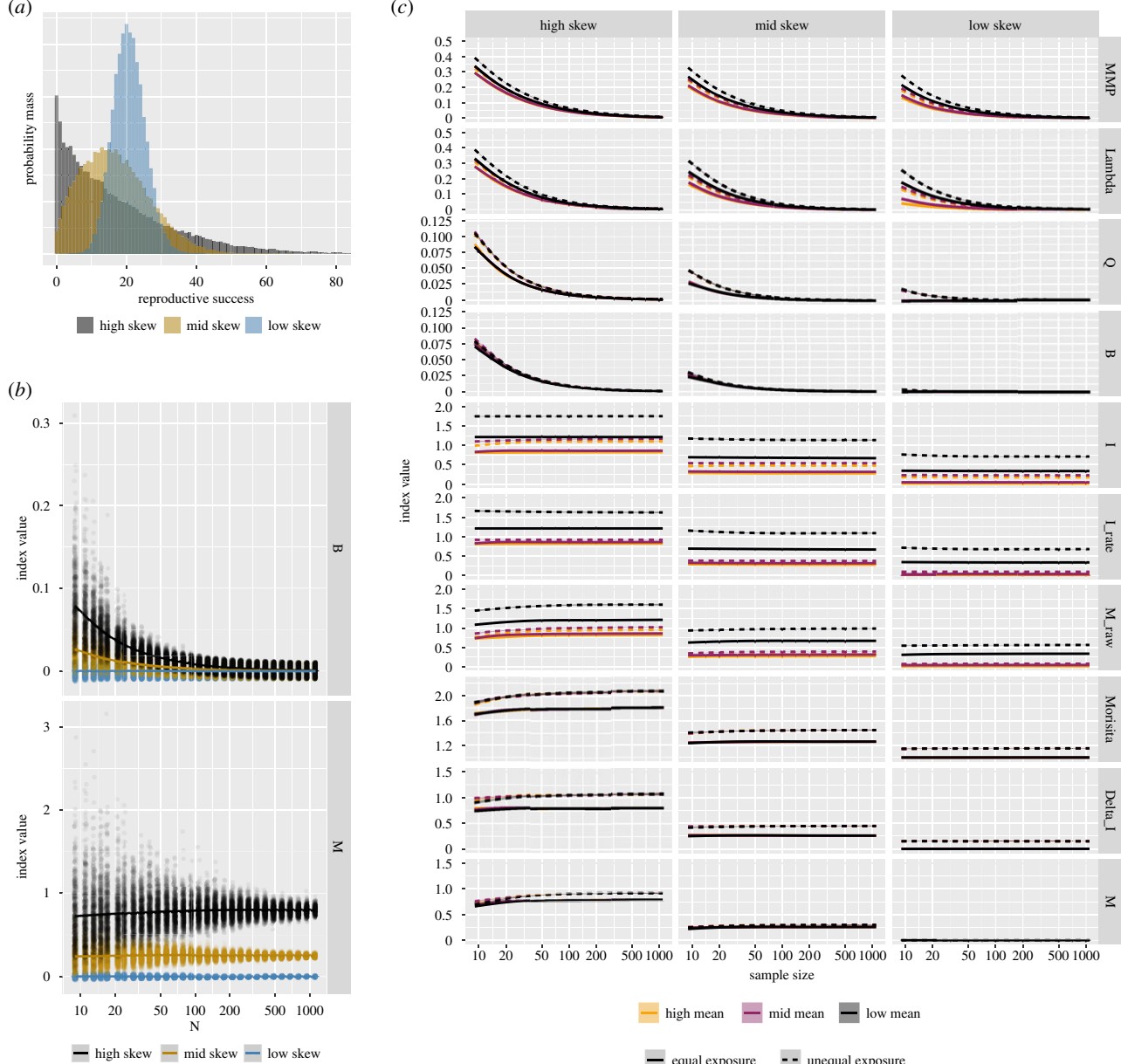

**Figure 1.** We analyse 270 000 simulated RS datasets to illustrate the effects of sample size (30 different levels), mean RS (three different levels), unequal exposure time (two different levels), and reproductive skew (three different levels) on 10 indices of reproductive skew. We consider three levels of skew for each combination of the other variables, and use 500 replicates per category combination to estimate a mean index value. In frame (*a*) we show the shape of three example Negative Binomial distributions from which RS realizations are drawn. Mean RS is constant across the plotted skew categories. In the high skew category (black), there is elevated probability mass on both low and high RS values, relative to the low skew category (blue). In the full simulation, we set the exponent of a rate scaling random effect (0.01 = low skew, 0.31 = mid skew, and 0.61 = high skew) to modulate skew as illustrated here. The mean of the Negative Binomial distribution was defined using a rate per exposure time unit of 1.0 = low, 7.0 = mid, and 20.0 = high. Sample size is modulated by randomly drawing *n* samples from each distribution. We then calculate each skew index using the *n*-vector of sample RS outcomes. We also consider the effect of variation in exposure time, with Equal Exposure time resulting from use of equal and fixed exposure times, and Unequal Exposure resulting from drawing exposure times from a uniform distribution. Frame (*b*) gives an example contrast between *B* and *M*. Mean RS is held constant within this frame. For each level of skew (low = blue, mid = yellow, high = black), we draw a random sample of *n* RS outcomes and then calculate *M* and *B* on this vector of simulated data. We repeat this process 500 times for each considered value of *n* and plot the results. The solid lines indicate average values. We see that—holding constant both mean RS *and* reproductive skew—*B* is highly sensitive to sample size. For large samples, *B* is actually insensitive to reproductive skew, and goes to zero regardless of the actual level of reproductive skew in the generative model; there is a structural bias in its mathematical definition. In contrast, *M* is sensitive to skew differences and is invariant to sample size. As sample size increases, *M* can cleanly differentiate between skew levels. We repeat this same analysis in frame (*c*) for all other combinations of variables and skew indices, but we plot only the mean trends. Specifically, we plot 'Maximum mating proportion' (MMP), $\lambda$, *Q*, *B*, *I*, *I* of RS rate, $\check{M}$, Morisita's $I_\sigma$, Waples' $\Delta_I$, and *M* as a function of sample size. The *x*-axis is plotted with log-transformed values, but labelled in natural units. Within each frame, the skew level is fixed. Colours are used to illustrate the effect of mean RS. Line-type is used to illustrate the effect of exposure time differences. A useful comparative measure of skew will: (i) be invariant to sample size (i.e. we should see flat horizontal lines within frames), (ii) be invariant to mean RS and exposure time changes (i.e. all lines should overlap within each frame), and (iii) be sensitive to skew (i.e. the *y*-axis locations of the lines should vary across columns). We observe: (1) MMP and $\lambda$ are sensitive to sample size, mean RS, and exposure time; (2) *Q* is invariant to mean RS, but not exposure time or sample size; (3) *B* is invariant to exposure time and mean RS, but varies sharply as a function of sample size; (4) while *I* and *I* of RS rate are invariant to sample size and can distinguish between levels of skew, they are sensitive to mean RS; (5) $\check{M}$, like *I*, is invariant to sample size, and can distinguish between levels of skew. It, however, remains sensitive to mean RS; (6) Morisita's $I_\sigma$ and Waples' $\Delta_I$ are invariant to mean RS and sample size, but remain sensitive to exposure time; finally, (7) *M* is largely invariant to sample size (except for small samples from highly skewed populations), invariant to mean RS, and largely invariant to exposure time (except in highly skewed populations, where it still outperforms Morisita's index and *I* of RS rate). (Online version in colour.)

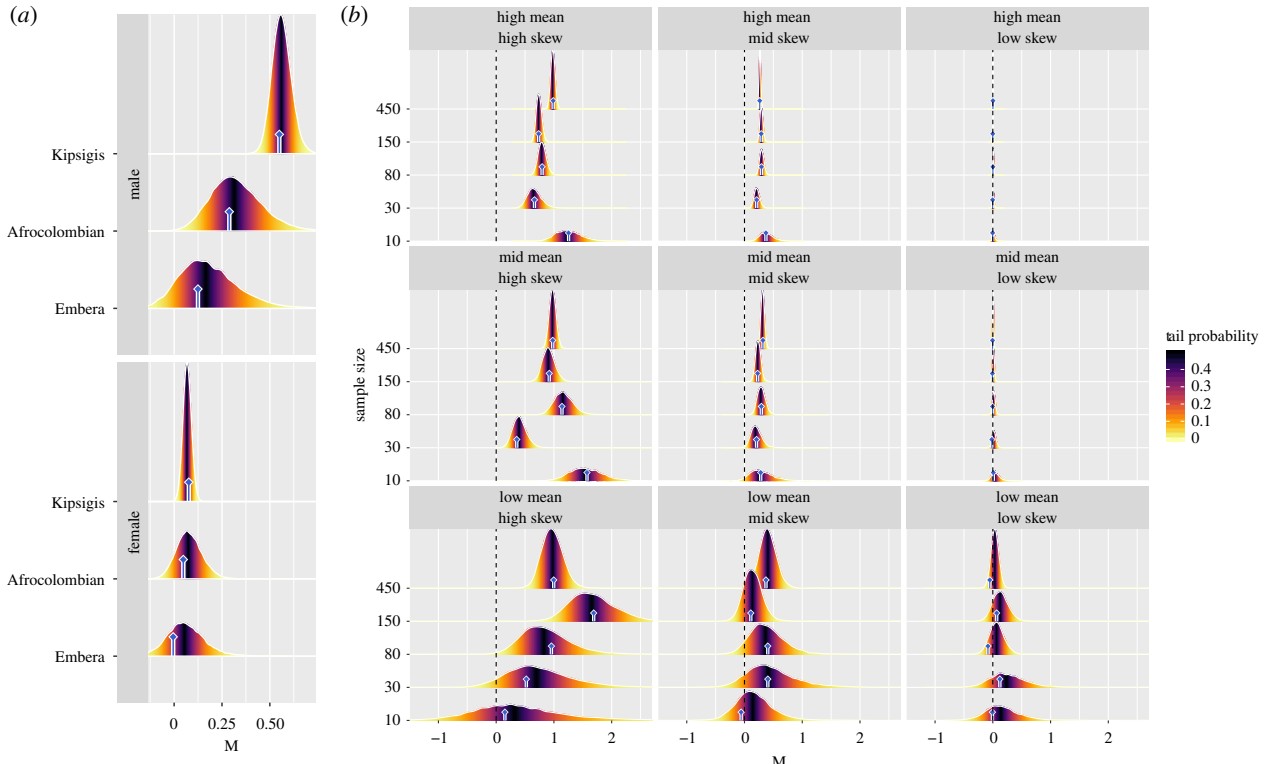

**Figure 2.** Frame (*a*): point estimates (blue bars) and posterior estimates (density distributions) of *M* for males and females in three human populations with different marriage systems. For males, *M* distinguishes the polygynous Kipsigis from the serially monogamous Afrocolombians—a mean difference of 0.23 (90% CI: 0.01, 0.46)—and the monogamous Emberá—a mean difference of 0.32 (90% CI: 0.07, 0.58). However, despite male Afrocolombians and Emberá having fairly distinct point estimates of *M*, a Bayesian approach suggests that it is hard to reliably conclude that there are skew differences between these populations given the relevant sample sizes—i.e. we see a mean difference of 0.08 (90% CI: −0.23, 0.42), where the posterior credible interval overlaps zero quite heavily. Among females, reproductive skew is approximately constant across populations, but the posterior estimate of *M* is most precise in the Kipsigis where population size is largest. Frame (*b*): posterior estimates of *M* for various simulated datasets, with various levels of skew and mean RS. Reproductive success was drawn randomly from a Negative Binomial distribution with a mean rate per exposure time unit of 1.0=low, 7.0=mid, and 20.0=high. To alter skew, we set the exponent of a rate scaling random effect to 0.01=low skew, 0.31=mid skew, and 0.61=high skew (as shown in figure 1*a*). Within each frame, we see the posterior distributions of *M* for various levels of sample size. Posterior estimates of *M* map closely onto the point estimates of *M*, plotted as blue bars. In general, we see that as the sample size increases, the posterior distributions narrow, reflecting more precise estimates of *M*. When the RS rate is low, there is necessarily less RS data and thus greater uncertainty in *M*, which is reflected in wider posterior credible regions. (Online version in colour.)

as marriage changes from polygynous, to serially monogamous, to monogamous—and that there can be substantial population-level uncertainty in *M*. This clarifies the importance of using Bayesian estimation methods to quantify and propagate the uncertainty inherent in a given skew estimate through all levels of analysis.

To investigate how posterior estimates of *M* are affected by sample size and RS rate, we simulate an array of datasets, and calculate posterior and point estimates of *M* from each. Figure 2(*b*) shows that the width of the posterior credible region shrinks with increasing sample size and RS rate. When sample size is small and/or RS events rare, skew is harder to measure and posterior credible regions are wider.

# 5. Relation to other measures

## (a) Standard variance measures

*M* is related to standard measures summarizing the second moment of a distribution. If all individuals have equal exposure time (i.e. $t_i = 1/N$ for all *i*), then:

$$\check{M} = \frac{\text{sd}^2}{\text{mean}^2} = \frac{\text{var}}{\text{mean}^2} = \text{cv}^2 = \frac{\varphi}{\text{mean}} = I, \tag{5.1}$$

where *sd* is the standard deviation, *var* is the variance, *cv* is the coefficient of variation, $\varphi$ is Crow and Morton's index of variability [72], and *I* is the opportunity for selection [10,40]. $\check{M}(r, t)$ is thus a generalization of *I(r)* to cases where individuals have unequal exposure time to risk of RS.

## (b) Nonacs' B index

Using our previous notation, *B* [35,36] is expressible as:

$$B(r, t) = \sum_{i=1}^{N} (\hat{r}_i - \hat{t}_i)^2 - \frac{N-1}{RN} \tag{5.2}$$

assuming that *N* in the second term on the right-hand side is the same as $\hat{N} = 1/\max(\hat{t})$ in Nonacs' formulation. This will hold approximately, as long as exposure time is not too unequal, since $1/\max(\hat{t})$ goes to *N* as $\hat{t}_i$ goes to $1/N$ for all *i*.

A key drawback of *B* for comparison across groups is that sample size, *N*, has a direct structural effect on *B*. To see why, note that terms $|\hat{r}_i - \hat{t}_i|$ will be of order $1/N$, so terms $(\hat{r}_i - \hat{t}_i)^2$ will be of order $1/N^2$. Assuming *R* and *N* are of the same order, the first term in *B* (and thus *B* itself) will be of order $1/N$. Analysis of *B* as a function of sample size, holding the underlying distribution of reproduction constant, confirms

that $B$ scales with the inverse of sample size (i.e. $B \sim 1/N$). This is shown in figure 1.

$M$ avoids this scaling issue while still allowing for unequal exposure times across individuals. Nevertheless, owing to their mathematical similarity, $\check{M}$ can be calculated from $B$ and *vice versa*:

$$\check{M}(r, t) = B(r, t)N + \frac{N-1}{R}. \tag{5.3}$$

## (c) Ruzzante's Q index

Ruzzante's [73] $Q$ index of relative monopolization is conceptually related to $B$ and $M$. If all individuals have equal exposure time (i.e. $t_i = 1/N$ for all $i$), then:

$$\check{M}(r, t) = \frac{N}{R}\left((R-1)Q(r) + 1\right). \tag{5.4}$$

## (d) Morisita's $I_\sigma$ index

Morisita's [46,47] $I_\sigma$ index is a sampling corrected version of $I$, and is expressible as:

$$I_\sigma(r) = N \frac{\left(\sum_{i=1}^{N} r_i^2\right) - R}{R(R-1)} = \frac{\mathrm{var}(r)N^2 + R^2 - RN}{R(R-1)}. \tag{5.5}$$

If all individuals have equal exposure time (i.e. $t_i = 1/N$ for all $i$), we can write $M(r, t)$ in closed form as: $M(r, t) = \frac{N^2}{R^2}\mathrm{var}(r) - \frac{N-1}{R}$. So, we have:

$$M(r, t) = \frac{(R-1)}{R}\left(I_\sigma(r) - 1\right) \tag{5.6}$$

and for large $R$, $I_\sigma$ approaches equivalence to $M(r, t)$ up to an additive constant.

## (e) Waples' $\Delta_I$ index

In work conceptually related to that by Morisita, Waples [48] uses a population genetics approach to derive a measure of skew, $\Delta_I(r)$, based on a similar correction of $I$:

$$\Delta_I(r) = I(r) - \mathbb{E}[I_{\mathrm{drift}}(r)] = I(r) - \frac{N-1}{R}. \tag{5.7}$$

For large $R$, $I_\sigma$, and $\Delta_I$ are equivalent up to an additive constant. When there is no age-structure or exposure time variation across individuals, $M$ is also equivalent to $\Delta_I$, since under equal exposure time: $M(r, t) = I(r) - \frac{N-1}{R}$. $M$ is thus a formal generalization of both $I_\sigma(r)$ and $\Delta_I(r)$ to cases where exposure time is variable across individuals.

## (f) Gini coefficient

The Gini coefficient [38,39] of a variable with $N$ observations is half of the relative mean distance between observations. So, if $t_i = 1/N$ for all $i$, the Gini coefficient of interest is:

$$\mathrm{Gini}(r) = \frac{\sum_{i=1}^{N}\sum_{j=1}^{N}|r_i - r_j|}{2RN}. \tag{5.8}$$

Continuing with this assumption, $\check{M}(r, t)$ can be written as:

$$\check{M}(r, t) = \frac{\sum_{i=1}^{N}\sum_{j=1}^{N}(r_i - r_j)^2}{2RN}. \tag{5.9}$$

Comparing these expressions, we see that $\check{M}$ is similar to the Gini coefficient, but uses squares instead of absolute values.

# 6. Re-evaluating comparative tests of skew using the multinomial index

$M$ does not decrease structurally with sample/group size as do other skew/inequality measures, such as $\lambda$ [74], the 'maximum mating proportion' (MMP) [8], and $B$ [36]. This raises a question as to whether previously observed associations between group size and reproductive skew represent biologically meaningful phenomena or statistical artefacts. To find out, we conduct a reanalysis of reproductive skew across populations using data from Kutsukake & Nunn [8]—henceforth K&N.

K&N report male group size as the only reliable predictor of mating skew among male primates. We took three steps to re-evaluate K&N's results. First, we examined univariate associations between several skew measures—$\lambda$, MMP, $B$, and $M$—with various demographic and reproductive variables (Table 1 of K&N). Second, we repeated their multiple regression analysis (Table 2 of K&N). Third, we repeated their intra-specific analysis of skew in chimpanzees. Finally, we developed a phylogenetically controlled mixed-effects model [75,76]. This last model: (i) increases power by using all data points instead of using species averages, (ii) estimates and controls for intra-specific variation through species-level random effects, (iii) estimates within- and between-species effects of each predictor simultaneously, and (iv) uses Bayesian methods to integrate over the uncertainty inherent in missing data observations, rather than dropping rows with missing covariates.

All data were taken from K&N's supplementary materials, and $M$ was calculated from $B$ (as described in §5b). Our statistical work-flow would be improved by estimating posterior distributions of $M$ using individual-level observations and then modelling this uncertainty in $M$ as measurement error in the comparative analysis; such individual-level data, however, are not yet available. A consensus phylogeny for all species in the sample was downloaded from the 10ktrees website Version 3 [77]. For steps 1 and 2, we followed K&N in setting all branch lengths equal (to 1) and log-transforming the data to meet the assumptions of their independent contrast analyses; however, since $B$ is sometimes negative, we did not log transform it or $M$. Independent contrast analyses were run using the `crunch` and `brunch` functions in the `caper` package [78] implemented in R [79]. Instead of identifying and excluding outlier contrasts by hand we relied on the built-in 'robust' argument.

For the phylogenetically controlled mixed-effects models, we used within-species centring by subtracting the species mean from each data point [75]; the model thus includes two slope estimates for each predictor, one for the species mean, and one for the group-level offset from the species mean. The model structure and fit diagnostics are described in detail in the electronic supplementary material, section 7.

## (a) Basic reanalysis

Sections 8.1 and 8.2 of the electronic supplementary material present the univariate and multivariate associations between various skew measures and demographic variables. MMP and $B$ show significant negative relationships with group size. When restricting the sample in the multivariate analysis to species for which all skew indices were available, all measures except $M$ show a significantly negative effect of male group size on skew.

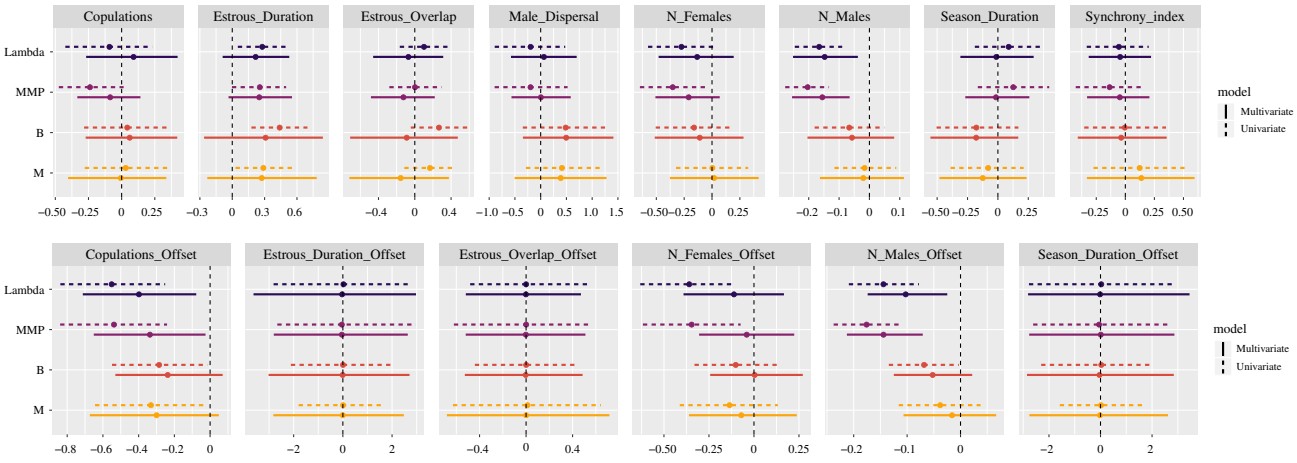

**Figure 3.** Posterior estimates of the effects of various predictor variables on reproductive skew using phylogenetically controlled mixed-effects models. Models were fit independently for each skew measure. In univariate models, each predictor was included in isolation from others. In the multivariate models, all predictors were included at the same time. Each bar plots the 90% posterior credible interval for the indicated effect, and each point plots the corresponding posterior mean. Effects that do not overlap the value of zero (dashed vertical lines) are credible at the 0.05 level. The top frame plots between-species effects, and the bottom frame plots within-species effects (i.e. the effect of group-level offsets from species-mean values). For any given predictor, both the species-level and group-level slopes were estimated in the same model. (Online version in colour.)

K&N also reported step-wise multiple regression models showing significant relationships between male group size and both $\lambda$ and MMP, and no relationships between the suite of considered skew measures and female group size or expected estrous overlap. We were able to reproduce these results; moreover, our analysis finds no significant relationship between $M$ and male group size, female group size, or expected estrous overlap, in comparable models. See electronic supplementary material, section 8.3 for more details.

## (b) Phylogenetic mixed-effects models

Phylogenetic methods and Markov Chain Monte Carlo software have significantly improved in the years since the original analysis by K&N; we use phylogenetically controlled mixed-effect models [75] implemented in Stan [80] via `brms` [81]. We extend the basic `Stan` models generated with `brms` to deal with missing data. This allows us to make use of the full content of the K&N dataset ($n = 84$ groups) and eliminates the need to drop rows with missing predictors. Additionally, we were able to use the full information content of the phylogenetic tree by using numerical branch lengths.

We fit a set of nine phylogenetic models for each outcome variable: $\lambda$, MMP, $B$, and $M$—one model was a multivariate model with the full set of predictors, and the other eight models included only a single predictor variable each—a robustness check. Figure 3 summarizes the results of the full set of these models.

In general, we find: (1) across skew indices, most predictor variables are uncorrelated with observed male reproductive skew. This is true for predictor variables expressed as species-specific means (i.e. between-species effects; figure 3, top frame) as well as for predictor variables expressed as group-specific offsets from species-specific means (i.e. within-species effects; figure 3, bottom frame). (2) MMP, $\lambda$— and sometimes $B$—detect male group size as a key predictor of male reproductive skew, at the species level and the group-offset level, in univariate and multivariate models. $M$, however, is not predicted by male group size. (3) MMP and $\lambda$ detect female group size as a key predictor of male reproductive skew, at both the species level and the group-

offset level, but only in univariate models. $M$ and $B$, however, are not predicted by female group size. (4) At the between-species level, all skew indices identify estrous duration as a positive correlate of male reproductive skew, but the reliability of the effect is attenuated when all predictors are included in a multivariate model. Finally, (5) at the within-species level, in both univariate and multivariate models, all skew metrics suggest that higher copulation rates are associated with lower levels of male reproductive skew.

We show that $M$ is capable of detecting skew and is associated with some of the same predictors as $\lambda$, MMP, and $B$. However, since $M$ is not structurally biased by group or sample size, it allows for better comparisons of skew in cross-species, cross-group, and cross-cultural contexts.

## (c) Summary

Our re-evaluation of the study by K&N suggests that the reported negative effect of male group size on male mating skew in primates does not persist when $M$ is used to measure reproductive skew. In this comparative study of male primates, sample size per group varied from $n = 2$ to $n = 10$ in 95% of the data, with two cases each of $n = 11$ and $n = 19$. Because of this, $M$ and $B$ are highly correlated in this set of data. Even so, as can be seen from the simulations in figure 1, $B$ is so sensitive to changes in sample size for small $n$, that even over this limited range, the adjustment from $B$ to $M$ removes the apparent effect of male group size on reproductive skew. In other comparative studies, with larger ranges of sample size per group, we would expect inferential contrasts between $M$ and other skew measures to be even larger.

## 7. Conclusions

The question of how to effectively measure skew/inequality, both within and among species, is emerging as an important and largely under-theorized question. Biologists [34] and economists [39,82] have struggled to develop inequality indices that are reliable and robust in comparative contexts. In this paper, we have addressed this issue by deriving and validating a new measure of reproductive skew that permits wider comparative research.

Proc. R. Soc. B 287: 20202025

royalsocietypublishing.org/journal/rspb　Proc. R. Soc. B 287: 20202025

Despite the long-standing interest in reproductive skew, many questions remain unanswered. The measure we have presented and analysed here—the multinomial index, $M$—provides a means of quantifying reproductive skew that avoids many of the issues affecting existing indices. In particular, $M$ is insensitive to variation in sample or group size, mean RS, age-structure, and even diminishing or highly nonlinear RS returns to age. It is also amenable to conversion to and from other common measures of skew and inequality, including Nonac's $B$, while remaining robust in the face of variation in the factors just listed.

To demonstrate the value of our index for cross-population comparisons, we have presented a series of simulation checks along with reanalyses of previously published comparative data on reproductive skew in male primates. The results indicate that some important empirical findings do not replicate when statistical biases in existing skew measures are analytically eliminated.

The multinomial index should prove useful in future comparative research on reproductive skew and other forms of inequality, such as mating access. To facilitate such work, we provide an R package, SkewCalc, for estimating $M$ from empirical data. We anticipate that future analyses employing the multinomial index will allow broader and more robust tests of theoretical models linking reproductive inequality and its causes and consequences.

Ethics. Regarding the Colombian data, informed consent was obtained from each respondent and the community leader (when appropriate) prior to data collection. Because of limited literacy rates at the study sites, informed consent was obtained verbally. All field protocols were approved by the Max Planck Institute for Evolutionary Anthropology, Department of Human Behavior, Ecology and Culture, and declared exempt from additional IRB oversight. Regarding the Kipsigis data, permission to conduct research in Kenya was granted by the Office of the President, Nairobi. Informed consent was obtained from each respondent. All field protocols were approved at Northwestern University. Special thanks to the Kipsigis families for their friendship and cheerful cooperation throughout the study.

Data accessibility. All data and code needed to replicate our findings are available at https://github.com/ctross/multinomialindex and the SkewCalc package is available at https://github.com/ctross/SkewCalc.

Authors' contributions. C.T.R. built the R package, designed the statistical analyses, and contributed to the derivation of the skew indices, especially the nonlinear age-adjustment methods. A.V.J. designed the reanalyses of published data and contributed to statistical analyses. P.L.H. and S.G. conceived of and derived the new skew indices. P.L.H., M.B.M., E.A.S., J.E.S., and S.G. conceived study and reviewed literature. All authors wrote and edited the manuscript.

Competing interests. We declare we have no competing interests.

Funding. This project was sponsored by the National Institute for Mathematical and Biological Synthesis, supported through National Science Foundation awards EF-0832858 and DBI-1300426, with additional support from The University of Tennessee, Knoxville. CR was supported by the Dynamics of Wealth Inequality project of the Behavioral Sciences Program at the Santa Fe Institute, the United States National Science Foundation (NSF-IBSS grant no. 1329089), and the Max Planck Institute for Evolutionary Anthropology. SG was supported by the U.S. Army Research Office grants W911NF-14-1-0637 and W911NF-17-1-0150, the Office of Naval Research grant W911NF-18-1-0138.

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
