## [Reviewer comments · Proceedings of the Royal Society B: Biological Sciences]

Review History

RSPB-2019-2261.R0 (Original submission)

Review form: Reviewer 1 (Peter Nonacs)

Recommendation

Major revision is needed (please make suggestions in comments)

Scientific importance: Is the manuscript an original and important contribution to its field?

Good

General interest: Is the paper of sufficient general interest?

Good

Quality of the paper: Is the overall quality of the paper suitable?

Acceptable

Is the length of the paper justified?

Yes

Should the paper be seen by a specialist statistical reviewer?

No

Do you have any concerns about statistical analyses in this paper? If so, please specify them explicitly in your report.

No

It is a condition of publication that authors make their supporting data, code and materials available - either as supplementary material or hosted in an external repository. Please rate, if applicable, the supporting data on the following criteria.

Is it accessible?

Yes

Is it clear?

Yes

Is it adequate?

Yes

Do you have any ethical concerns with this paper?

No

Comments to the Author

See attached file.

Review form: Reviewer 2

Recommendation

Major revision is needed (please make suggestions in comments)

Scientific importance: Is the manuscript an original and important contribution to its field?

Good

General interest: Is the paper of sufficient general interest?

Good

Quality of the paper: Is the overall quality of the paper suitable?

Acceptable

Is the length of the paper justified?

No

Should the paper be seen by a specialist statistical reviewer?

Yes

Do you have any concerns about statistical analyses in this paper? If so, please specify them explicitly in your report.

Yes

It is a condition of publication that authors make their supporting data, code and materials available - either as supplementary material or hosted in an external repository. Please rate, if applicable, the supporting data on the following criteria.

Is it accessible?

Yes

Is it clear?

No

Is it adequate?

Yes

Do you have any ethical concerns with this paper?

Yes

Comments to the Author

Dear authors,

A new measure of reproductive skew which controls for differences in group size between samples is a valuable contribution to the field of social evolution and behavioural ecology, which is why I find the manuscript of sufficient general interest.

Though I suggest the following changes before publication.

The introduction of the manuscript and the introduction of the new index is well written and comprehensible. In the introduction of the new index (section 2. A comparable measure of skew, line 105 - 148), "X" and "E" in equation 2.1 as well as the equation 2.3 is not further explained. In section 3 (Derivation) a long list of equations is listed which are not further explained and hence it is not clear of what use they are to understand the manuscript. This needs to be further explained/ mentioned or I suggest excluding some of the equations.

In section 4, the new reproductive skew index gets set into relation to other skew measures. Here, I do not see what extra information is added by comparing the multinomial index, M , to more than the Nonacs' binomial index. Especially because the other skew measures (Coefficient of variation, opportunity for Selection and Gini coefficient) are not further mentioned in the manuscript. This needs to be further explained/ mentioned or I suggest excluding all skew measures besides the B index for comparison with the multinomial index.

In section 5, data from Kutsukake and Nunn (2006) is re-evaluated with the multinomial index. Here another set of skew measures used in Kutsukake and Nunn (2006) is compared with the new index. It might be useful to compare these skew measures (λ and maximum mating proportion) with the multinomial index in section 4.

In section 5a, b and c, information on the univariate associations between the skew measures, multiple regression analysis and on the step-wise multiple regression models are missing. Information on included predictors in models, used R packages, checked assumptions, and used statistical test should be at least stated in the supplementary online material and mentioned in the main manuscript text.

In the summary (section 5e) the authors state that the multinomial index and B index are highly correlated due to the small sample sizes in the Kutsukake and Nunn (2006) data set and expect larger differences between indices with data of larger sample size differences. Sample sizes of $N=2$ to $N=10$ are not sufficient for statistical analysis these days and I suggest comparing it to published reproductive skew data with larger sample sizes (e.g. Stier et al. 2011, Engelhardt et al. 2017, Surbeck et al. 2017, Minkner et al. 2018).

Some general notes on the manuscript:

1. I am missing an ethical statement concerning the conducted human reproductive success research.
2. The use of two terms (reproductive skew and reproductive inequality) of similar meaning is confusing and I suggest using one of the terms throughout the manuscript.
3. Please check that you use introduced abbreviations all the time, e.g. MMP (line 404) as well as spell out SOM at least ones.
4. Concerning data availability, I suggest making more clear where to find the human populations reproductive success data (Kipsigis, Afrocolombians and Emberá) on the main authors GitHub.

Otherwise I enjoyed reading the manuscript and find it of high future value for comparisons between groups, populations and species research.

Kind regards

Decision letter (RSPB-2019-2261.R0)

07-Nov-2019

Dear Dr Ross:

I am writing to inform you that we have now obtained responses from referees on manuscript RSPB-2019-2261 entitled "The Multinomial Index: A Robust Measure of Reproductive Skew" which you submitted to Proceedings B.

Unfortunately, on the advice of the Associate Editor and the referees, your manuscript has been rejected following full peer review. Competition for space in Proceedings B is currently extremely severe, as many more manuscripts are submitted to us than we have space to print. We are therefore only able to publish those that are exceptional, convincing and present significant advances of broad interest, and must reject many good manuscripts.

Please find below the comments received from referees concerning your manuscript, not including confidential reports to the Editor. I hope you may find these useful should you wish to submit your manuscript elsewhere. While you will see that there is a consensus that elements of the topic are interesting, overall, the ratings, tone and nature of issues raised, collectively mean that we are unable to take your manuscript forward. When making such decisions, we consider carefully not only the nature of the referee reports, and associated ratings, but also confidential comments, and the reading of the manuscript by the Board Member and Editor. Additionally, our decisions are further informed by the baseline of quality expectations and criteria, so as to ensure appropriate benchmarking throughout our peer review process. I hope you may find these useful should you wish to submit your manuscript elsewhere.

We are sorry that your manuscript has had an unfavourable outcome, but would like to thank you for offering your work to Proceedings B.

Sincerely,
Professor Gary Carvalho
mailto:proceedingsb@royalsociety.org

Associate Editor
Board Member: 1
Comments to Author:

The authors provide a new measure of reproductive skew, which is not biased by samples size. I particularly like that the authors compare their measure to existing ones. Nevertheless I really had some difficulties reading the manuscript.

1. I found it confusing that the authors sometimes use reproductive skew and sometimes inequality.
2. Some of the abbreviations are explained in the text, but not used there (e.g. l 117. opportunity for selection I,). It only appears later in the figure.
3. Some are not explained, e.g. SOM. Although most readers probably know what that means, I think it is much easier to understand if explained the first time it is used.
4. is there a reason why some of the equations are in the text, while others have their own line?

Reviewer(s)' Comments to Author:

Referee: 1

Comments to the Author(s)
See attached file.

Referee: 2

Comments to the Author(s)

Dear authors,

A new measure of reproductive skew which controls for differences in group size between samples is a valuable contribution to the field of social evolution and behavioural ecology, which is why I find the manuscript of sufficient general interest.

Though I suggest the following changes before publication.

The introduction of the manuscript and the introduction of the new index is well written and comprehensible. In the introduction of the new index (section 2. A comparable measure of skew, line 105 - 148), "X" and "E" in equation 2.1 as well as the equation 2.3 is not further explained.

In section 3 (Derivation) a long list of equations is listed which are not further explained and hence it is not clear of what use they are to understand the manuscript. This needs to be further explained/mentioned or I suggest excluding some of the equations.

In section 4, the new reproductive skew index gets set into relation to other skew measures. Here, I do not see what extra information is added by comparing the multinomial index, M, to more than the Nonacs' binomial index. Especially because the other skew measures (Coefficient of variation, opportunity for Selection and Gini coefficient) are not further mentioned in the manuscript. This needs to be further explained/mentioned or I suggest excluding all skew measures besides the B index for comparison with the multinomial index.

In section 5, data from Kutsukake and Nunn (2006) is re-evaluated with the multinomial index. Here another set of skew measures used in Kutsukake and Nunn (2006) is compared with the new index. It might be useful to compare these skew measures (lambda and maximum mating proportion) with the multinomial index in section 4.

In section 5a, b and c, information on the univariate associations between the skew measures, multiple regression analysis and on the step-wise multiple regression models are missing. Information on included predictors in models, used R packages, checked assumptions, and used statistical test should be at least stated in the supplementary online material and mentioned in the main manuscript text.

In the summary (section 5e) the authors state that the multinomial index and B index are highly correlated due to the small sample sizes in the Kutsukake and Nunn (2006) data set and expect larger differences between indices with data of larger sample size differences. Sample sizes of N=2 to N=10 are not sufficient for statistical analysis these days and I suggest comparing it to published reproductive skew data with larger sample sizes (e.g. Stier et al. 2011, Engelhardt et al. 2017, Surbeck et al. 2017, Minkner et al. 2018).

Some general notes on the manuscript:

1. I am missing an ethical statement concerning the conducted human reproductive success research.
2. The use of two terms (reproductive skew and reproductive inequality) of similar meaning is confusing and I suggest using one of the terms throughout the manuscript.
3. Please check that you use introduced abbreviations all the time, e.g. MMP (line 404) as well as spell out SOM at least ones.
4. Concerning data availability, I suggest making more clear where to find the human populations reproductive success data (Kipsigis, Afrocolombians and Emberá) on the main authors GitHub.

Otherwise I enjoyed reading the manuscript and find it of high future value for comparisons between groups, populations and species research.

Kind regards

Author's Response to Decision Letter for (RSPB-2019-2261.R0)

See Appendix A.

RSPB-2020-0843.R0

Review form: Reviewer 2

Recommendation

Accept with minor revision (please list in comments)

Scientific importance: Is the manuscript an original and important contribution to its field?

Good

General interest: Is the paper of sufficient general interest?

Good

Quality of the paper: Is the overall quality of the paper suitable?

Good

Is the length of the paper justified?

Yes

Should the paper be seen by a specialist statistical reviewer?

Yes

Do you have any concerns about statistical analyses in this paper? If so, please specify them explicitly in your report.

Yes

It is a condition of publication that authors make their supporting data, code and materials available - either as supplementary material or hosted in an external repository. Please rate, if applicable, the supporting data on the following criteria.

Is it accessible?

Yes

Is it clear?

Yes

Is it adequate?

Yes

Do you have any ethical concerns with this paper?

No

Comments to the Author

Dear authors,

I appreciate that the suggested changes and concerns for this manuscript were addressed and the manuscript was revised where feasible.

The manuscript has improved and only minor changes are suggested.

I would advise to further improve the manuscript by stating at first mentioning of the indices (multinomial index, binomial index, opportunity for selection, 'uncorrected' multinomial index) the term which will be used throughout the manuscript. As for example, throughout the manuscript up to five different terms are used for the multinomial index (e.g. L87: multinomial index, M, L88: M, L113: multinomial index, L194: M index,) and the binomial index (e.g. L65: Nonacs' B index, L85: B, L193: binomial index, B, L338: Nonacs' B, L346: Nonacs' binomial index). It would make the manuscript easier to read.

Additionally, the use of the two terms skew and inequality were explained by the authors, which I appreciated. Though I suggest not using the two terms within the same sentence (L138-144).

Last but not least, please replace 'reproductive success' with RS in line 174.

Kind regards

Review form: Reviewer 3

Recommendation

Accept with minor revision (please list in comments)

Scientific importance: Is the manuscript an original and important contribution to its field?

Acceptable

General interest: Is the paper of sufficient general interest?

Acceptable

Quality of the paper: Is the overall quality of the paper suitable?

Marginal

Is the length of the paper justified?

No

Should the paper be seen by a specialist statistical reviewer?

Yes

Do you have any concerns about statistical analyses in this paper? If so, please specify them explicitly in your report.

No

It is a condition of publication that authors make their supporting data, code and materials available - either as supplementary material or hosted in an external repository. Please rate, if applicable, the supporting data on the following criteria.

Is it accessible?

N/A

Is it clear?

N/A

Is it adequate?

N/A

Do you have any ethical concerns with this paper?

No

Comments to the Author

In this paper authors develop a multinomial index of reproductive skew the main advantage of which is that it is not biased by differences in sample/group size, a problem with some existing indices, notably, the authors mention Nonacs Binomial Index B . The paper is well written and the subject is of relative importance. One concern lies in the degree to which this index, its assumptions and utility relates to an index of resource monopolization published over two decades ago, referred to as index Q and based on assumptions of multinomial distributions (Behavioral Ecology 1996 Vol. 7 No. 2: 199-207). How does the index of RS described in the present paper relate to the index Q introduced in that earlier study?

Review form: Reviewer 4 (Robin Waples)

Recommendation

Major revision is needed (please make suggestions in comments)

Scientific importance: Is the manuscript an original and important contribution to its field?

Excellent

General interest: Is the paper of sufficient general interest?

Good

Quality of the paper: Is the overall quality of the paper suitable?

Good

Is the length of the paper justified?

Yes

Should the paper be seen by a specialist statistical reviewer?

No

Do you have any concerns about statistical analyses in this paper? If so, please specify them explicitly in your report.

No

It is a condition of publication that authors make their supporting data, code and materials available - either as supplementary material or hosted in an external repository. Please rate, if applicable, the supporting data on the following criteria.

Is it accessible?

Yes

Is it clear?

N/A

Is it adequate?

N/A

Do you have any ethical concerns with this paper?

No

Comments to the Author

See attached file

Decision letter (RSPB-2020-0843.R0)

27-May-2020

I am writing to inform you that this version of your manuscript RSPB-2020-0843 entitled "The Multinomial Index: A Robust Measure of Reproductive Skew" has, in its current form, been rejected for publication in Proceedings B. This action has been taken on the advice of referees, who have recommended that substantial revisions are necessary. With this in mind we would be happy to consider a resubmission, provided the comments of the referees are fully addressed. However please note that this is not a provisional acceptance.

Please find below the comments made by the referees, not including confidential reports to the Editor, which I hope you will find useful. While you will see that there is a consensus that elements of the topic are interesting, overall, the ratings, tone and nature of issues raised (esp. those from referee 3), collectively mean that we are unable to take your manuscript forward in its current form. When making such decisions, we consider carefully not only the nature of the referee reports, and associated ratings, but also confidential comments, and the reading of the manuscript by the Editor. Additionally, our decisions are further informed by the baseline of quality expectations and criteria, so as to ensure appropriate benchmarking throughout our peer review process. If you do choose to resubmit your manuscript, please upload the following:

Sincerely,
Professor Gary Carvalho
mailto: proceedingsb@royalsociety.org

Associate Editor
Comments to Author:

We thank the authors for addressing some issues, and while there remain some substantive aspects for you to consider, we provide a final opportunity for you to revise your manuscript. Particularly the third referee provided very detailed comments, which will definitely help to further improve the manuscript, e.g. by clarifying if and how sampling at different life stages can influence the measurement.

Reviewer(s)' Comments to Author:

Referee: 3

Comments to the Author(s).

In this paper authors develop a multinomial index of reproductive skew the main advantage of which is that it is not biased by differences in sample/group size, a problem with some existing indices, notably, the authors mention Nonacs Binomial Index B . The paper is well written and the subject is of relative importance. One concern lies in the degree to which this index, its assumptions and utility relates to an index of resource monopolization published over two decades ago, referred to as index Q and based on assumptions of multinomial distributions (Behavioral Ecology 1996 Vol. 7 No. 2: 199-207). How does the index of RS described in the present paper relate to the index Q introduced in that earlier study?

Referee: 2

Comments to the Author(s).

Dear authors,

I appreciate that the suggested changes and concerns for this manuscript were addressed and the manuscript was revised where feasible.

The manuscript has improved and only minor changes are suggested.

I would advise to further improve the manuscript by stating at first mentioning of the indices (multinomial index, binomial index, opportunity for selection, 'uncorrected' multinomial index) the term which will be used throughout the manuscript. As for example, throughout the manuscript up to five different terms are used for the multinomial index (e.g. L87: multinomial index, M, L88: M, L113: multinomial index, L194: M index,) and the binomial index (e.g. L65: Nonacs' B index, L85: B, L193: binomial index, B, L338: Nonacs' B, L346: Nonacs' binomial index). It would make the manuscript easier to read.

Additionally, the use of the two terms skew and inequality were explained by the authors, which I appreciated. Though I suggest not using the two terms within the same sentence (L138-144).

Last but not least, please replace 'reproductive success' with RS in line 174.

Kind regards

Referee: 4

Comments to the Author(s).

See attached file

Author's Response to Decision Letter for (RSPB-2020-0843.R0)

See Appendix B.

RSPB-2020-2025.R0

Review form: Reviewer 4 (Robin Waples)

Recommendation

Accept with minor revision (please list in comments)

Scientific importance: Is the manuscript an original and important contribution to its field?

Excellent

General interest: Is the paper of sufficient general interest?

Good

Quality of the paper: Is the overall quality of the paper suitable?

Excellent

Is the length of the paper justified?

Yes

Should the paper be seen by a specialist statistical reviewer?

No

Do you have any concerns about statistical analyses in this paper? If so, please specify them explicitly in your report.

No

It is a condition of publication that authors make their supporting data, code and materials available - either as supplementary material or hosted in an external repository. Please rate, if applicable, the supporting data on the following criteria.

Is it accessible?

No

Is it clear?

N/A

Is it adequate?

N/A

Do you have any ethical concerns with this paper?

No

Comments to the Author

The authors provided a detailed response to my comments on the previous version, and the revised manuscript is much improved. In particular, since I deal with similar issues from a population genetics perspective, it is easier to understand the anthropological perspective in the revised version; the previous version was skimpy on those details and basically assumed that the anthropological perspective is the only way to think about these issues. That should make the manuscript more accessible to the general reader. If I wanted to quibble, I might point out that although the existence of other perspectives is acknowledged in the revised version, the population genetics perspective is not really developed in any meaningful way.

My bioRxiv paper on the delta I index is now published in Evolution (available at

<http://dx.doi.org/10.1111/evo.14061>), so that citation can be updated. All the major results and conclusions remain unchanged, but the published version contains a couple of additions that the authors might look at to see whether there is anything useful relative to their paper on the multinomial index.

1) In response to reviewer/editor comments about notation, I redid the analytical model in terms of a generalized Wright-Fisher model of reproduction. In the standard WF model, each parent contributes equally to a large (~infinite) pool of gametes that unite at random to form the offspring generation. This can be explicitly modeled in a computer by allowing each offspring to 'choose' its parents randomly and with replacement from the pool of adults, all of whom have an equal probability of being selected at each draw. The generalized WF model differs in allowing unequal contributions to the initial gamete pool, which can be modeled by allowing different adults to have different probabilities of being a parent. These probabilities can be expressed as a vector of weights, W . The potential relevance of this is that the expected value of Crow's I and ΔI can be expressed as simple functions of the squared CV of W .

2) Like the authors, the major focus in my paper was on finding a way to adjust for effects of mean offspring number in a sample, which can reflect experimental design, sampling effort, logistical constraints, and other factors that represent noise. However, a colleague of mine who works with fantastic long-term datasets on reproductive success of great tits provided a novel perspective: in their studies, they basically inventory the entire population every year, but sample size of recruits varies across years as a result of demographic and environmental fluctuations. In this situation, temporal variation in mean offspring number reflects temporal changes in mean fitness in the population as a whole, not just in the sample. In this special case, therefore, removing this effect using the ΔI index could be obscuring a true biological signal related to intensity of selection. I added a short para near the end of Discussion pointing out this issue, and the authors might consider whether this issue is also potentially relevant to the M index.

L 61: "One desiderata" -> one desideratum

Robin Waples

Decision letter (RSPB-2020-2025.R0)

07-Sep-2020

Dear Dr Ross

I am pleased to inform you that your manuscript RSPB-2020-2025 entitled "The Multinomial Index: A Robust Measure of Reproductive Skew" has been accepted for publication in Proceedings B.

The referee(s) have recommended publication, but also suggest some minor revisions to your manuscript. Therefore, I invite you to respond to the referee(s)' comments and revise your manuscript. Because the schedule for publication is very tight, it is a condition of publication that you submit the revised version of your manuscript within 7 days. If you do not think you will be able to meet this date please let us know.

[http://datadryad.org/submit?journalID=RSPB&manu=\(Document not available\)](http://datadryad.org/submit?journalID=RSPB&manu=(Document not available)) which will

take you to your unique entry in the Dryad repository. If you have already submitted your data to dryad you can make any necessary revisions to your dataset by following the above link. Please see <https://royalsociety.org/journals/ethics-policies/data-sharing-mining/> for more details.

Sincerely,
Professor Gary Carvalho
mailto:proceedingsb@royalsociety.org

Associate Editor

Comments to Author:

I think the authors made a good job in further improving the paper. I only have a minor thing.

l 201 - check formatting

Reviewer(s)' Comments to Author:

Referee: 4

Comments to the Author(s).

The authors provided a detailed response to my comments on the previous version, and the revised manuscript is much improved. In particular, since I deal with similar issues from a population genetics perspective, it is easier to understand the anthropological perspective in the revised version; the previous version was skimpy on those details and basically assumed that the anthropological perspective is the only way to think about these issues. That should make the manuscript more accessible to the general reader. If I wanted to quibble, I might point out that although the existence of other perspectives is acknowledged in the revised version, the population genetics perspective is not really developed in any meaningful way.

My bioRxiv paper on the delta I index is now published in *Evolution* (available at <http://dx.doi.org/10.1111/evo.14061>), so that citation can be updated. All the major results and conclusions remain unchanged, but the published version contains a couple of additions that the authors might look at to see whether there is anything useful relative to their paper on the multinomial index.

1) In response to reviewer/editor comments about notation, I redid the analytical model in terms of a generalized Wright-Fisher model of reproduction. In the standard WF model, each parent contributes equally to a large (~infinite) pool of gametes that unite at random to form the offspring generation. This can be explicitly modeled in a computer by allowing each offspring to 'choose' its parents randomly and with replacement from the pool of adults, all of whom have an equal probability of being selected at each draw. The generalized WF model differs in allowing unequal contributions to the initial gamete pool, which can be modeled by allowing different adults to have different probabilities of being a parent. These probabilities can be expressed as a vector of weights, W . The potential relevance of this is that the expected value of Crow's I and delta I can be expressed as simple functions of the squared CV of W .

2) Like the authors, the major focus in my paper was on finding a way to adjust for effects of mean offspring number in a sample, which can reflect experimental design, sampling effort, logistical constraints, and other factors that represent noise. However, a colleague of mine who works with fantastic long-term datasets on reproductive success of great tits provided a novel perspective: in their studies, they basically inventory the entire population every year, but sample size of recruits varies across years as a result of demographic and environmental

fluctuations. In this situation, temporal variation in mean offspring number reflects temporal changes in mean fitness in the population as a whole, not just in the sample. In this special case, therefore, removing this effect using the delta I index could be obscuring a true biological signal related to intensity of selection. I added a short para near the end of Discussion pointing out this issue, and the authors might consider whether this issue is also potentially relevant to the M index.

L 61: "One desiderata" -> one desideratum

Robin Waples

Author's Response to Decision Letter for (RSPB-2020-2025.R0)

See Appendix C.

Decision letter (RSPB-2020-2025.R1)

14-Sep-2020

Dear Dr Ross

I am pleased to inform you that your manuscript entitled "The Multinomial Index: A Robust Measure of Reproductive Skew" has been accepted for publication in Proceedings B.

Open Access

Paper charges

Sincerely,
Proceedings B
<mailto:proceedingsb@royalsociety.org>

Appendix A

Dear Dr Ross,

Your appeal has now been considered by the Editor. I am pleased to let you know that on this occasion, the Editor has decided to allow your appeal, and invites you to resubmit your manuscript to the journal. Specific comments from the Editor are included below:

****Editor comments****

Thank you for taking the time and for your constructive approach to the above rejected manuscript. On this occasion, I am happy to provide an opportunity for revision, and resubmission. We will of course decide ourselves how to proceed with selecting appropriate referees, but I accept the potential conflict-of-interest that you identify. Notwithstanding: please understand the following 2 key points, should you decide to resubmit. First, that the offer to resubmit, does not of course, as with any peer review process, guarantee eventual publication. 2nd, that while I can see the potential conflict-of-interest from one of the referees, overall, the ratings and confidential feedback, indicated a lack of sufficient general interest, as well as likely impact, based on our benchmarking across a wide number of manuscripts. Therefore, in your resubmission, do whatever is possible, to increase potential impact, through enhancing in an explicit manner, how this work significantly advances the field, with an increased likelihood for uptake and more wide-scale applicability.

Dear editor, we thank you for giving us the opportunity on appeal to respond to the referees and make our case. We argue that this paper introduces an important new metric for measuring reproductive skew that: 1) is rigorously derived and then quality-checked under a large range of simulated and empirical data sets, 2) will significantly broaden the empirical literature on quantitative models of skew, relative to the metrics currently in widest use, by permitting statistically rigorous cross-group comparisons for the first time, and 3) will advance scientific theory on the causes and consequences of reproductive inequality/skew.

To address point (1), we have responded to the reviewers' objections to our metric's performance. We show in our paper that the most widely used metric of skew at the current time, Nonacs' B, has a fundamental structural bias based on sample/group size that has confounded all comparative studies based on it for the last 20 years. This bias is very likely to have impacted theory building in the field, as the effect of the bias term in B is substantial (B scales with $1/N$), where N is group/sample size. This bias can easily mask any other effect. Accordingly, we believe that M is thus an essential improvement on B for doing any comparative work on skew. We provide a short 3 sentence proof of the scaling relationship of B with sample size in lines 355-365, along with a large scale simulation of B as a function of sample size in Figure 1.

Similarly, the same reviewer voices other objections to our methods without proof or mathematical defence of such objections (i.e., stating it "is unclear how M values are actually generated" or "M simply exchanges one bias for another"). In rebuttal, we refer to evidence to the contrary in our manuscript (i.e., an explicit definition of M [Eqs. 3.1-3.3], and robustness checks showing no statistical biases over a large array of possible input distributions for RS, age, and sample size [Figure 1]). In other cases, we provide a direct rebuttal only in this letter, as it makes little sense to include these debates in a public forum (e.g., in the case where the reviewer incorrectly claims we use non-standard definitions of the words *sample* and *population*). However, whenever possible, we have tried to read the reviewers' comments in the most positive light, with the awareness that we can use any misunderstandings uncovered at this point of the review process to clarify our language, strengthen our arguments, and minimize the probability that other readers will misunderstand similar points. In cases where the reviewers raised solid points, we have modified the main text and indicated the locations of such changes with line numbers here.

To address point (2) we have tried to do a better job at explaining why M is empirically needed. In the initial submission, we—admittedly—did a sub-par job of linking the usefulness of M to the field of evolutionary biology more widely. We have significantly revised several paragraphs in the introduction to emphasize that M allows us to test comparative (i.e., cross-population or cross-

species) models and predictions about the social and ecological causes and consequences of reproductive skew that are simply not possible to test with current metrics like B or Morisita's index (lines 60-70). We also show that empirical studies using the most widely used skew measures reach incorrect conclusions due to the mathematical biases present in these existing skew measures (Section 6). This should also help to establish the broader impact of this paper.

We address point (3) in two ways. First, in this letter, we note that studies on, and measures of, reproductive skew have generally been of wide interest: Hanna Kokko's work on the λ index, Peter Nonacs' work on the B index, James Crow's work on the I index (the opportunity for selection), and Morisita's work on the I_σ index each have hundreds of citations in Google Scholar (in the case of Crow and Morisita, more than a thousand). However, as we show in Figure 1 and its caption, each of these measures is affected by at least one of three important problems: mean dependency, sample size dependency, or age-structure dependency. Since these are three of the biggest issues preventing the uptake of high-impact, cross-group, comparative research, we set out to derive a more general skew metric that is simultaneously robust to each of them.

In the caption to Figure 1, we describe how the λ index is sensitive to group/sample size, age structure, and mean RS, and thus isn't well suited for comparative work. The B index thoroughly addresses the issue with age-structure, but remains highly sensitive to sample/group size, and thus isn't well suited for comparative work either. The I-index deals reasonably well with variation in mean RS, but is sensitive to age-structure. The M index, in contrast to all others metrics, is invariant to each of these issues.

As Peter Nonacs points out below in response to our manuscript, current measures of skew come with severe limitations on inference in cross-group contexts: being limited essentially to testing whether or not the pattern of skew observed in a population is explainable by random processes. However, as a field, evolutionary scientists are moving away from simple, group-specific, null-hypothesis significance testing about "skew" or "no skew". We now have many powerful theoretical models (e.g., see lines 21-36 and 120-138 in our main text) describing dynamic, continuous relationships between ecological context, marriage and mating systems, and reproductive skew. We review this work in our introduction. Empirical tests of such models, however, require cross-group comparative research on the relationship between group-level social/ecological variables and group-level skew, and this itself requires a skew metric that allows for such comparisons. None of the extant metrics allow for rigorous comparative research; M does. We demonstrate this claim using analysis of 270,000 simulated data sets with 8 different skew measures in Figure 1. As such, we strongly believe that M will be of wide interest to the field, and will help stimulate many comparative publications about the linkages between reproductive inequality and social evolution.

We also note here that we did not set out to simply make up a new skew measure, or to pick apart a metric derived by another researcher. Rather, we—as the organizers of a team of co-authors composed of more than 70 evolutionary biologists and anthropologists—have compiled a data set of individual-level reproductive outcomes from thousands of people across 97 small-scale human populations along with RS data from 76 species of non-human mammals, with the purpose of understanding the causes and consequences of reproductive skew in comparative perspective (this is the largest data set compiled for such a purpose that we are aware of). We are preparing this empirical paper for publication, but think it is important to publish the theoretical basis for it in advance, and validate our skew measure through simulation, as done in the present submission.

Our empirical work is being revised for resubmission at PNAS, and one of the key comments from reviewers there was that we needed to submit our theoretical and methodological contribution on the M index for peer review independently as a separate paper. They argued that we needed to validate M using simulated data, contrasting it to the best available skew metrics, prior to using it on empirical data. This is what inspired us to write the current paper introducing and validating the M metric.

Second, as we acknowledged above, the editor is right that we did not stress the potential impact of this paper strongly enough in the text of our manuscript itself. In our initial submission, we tried to be

as delicate as possible in our critique of existing skew metrics. We did not want to be seen as picking on Nonacs' work: we were, in fact, quite inspired by Nonacs' approach to dealing with exposure time. However, this worry has possibly led to us under-stating the potential impact and scientific importance of our new metric. In our revised introduction, we have attempted to remain cordial and positive about the incremental progress contributed by past work on different measures of skew, while also emphasizing the much broader scope for research on skew through the use of M . See especially lines 75-115, where we introduce the flow of ideas in the paper, and the motivations for each step taken.

Associate Editor

Board Member: 1

Comments to Author:

The authors provide a new measure of reproductive skew, which is not biased by sample size.

I particularly like that the authors compare their measure to existing ones.

Nevertheless I really had some difficulties reading the manuscript.

1. I found it confusing that the authors sometimes use reproductive skew and sometimes inequality.

We now define "skew" as a measure of reproductive inequality in both the abstract and introductory sentence (lines 2-4). We include a footnote on line 3 to explain our word choice. Reproductive skew and reproductive inequality are essentially synonymous terms with different usage frequencies in different sub-fields of evolutionary biology. We want to speak to as broad an audience as possible, so we have tried to use both terms throughout. For clarity, we now use "skew/inequality" in key places.

2. Some of the abbreviations are explained in the text, but not used there (e.g. 1 117. opportunity for selection I ,). It only appears later in the figure.

On the old line 117 (now line 186), we introduced the key point that our M metric is a generalization of the I -index to cases where exposure time to risk of RS (i.e., age, or time under observation) is variable across units, but we did so in an awkward place.

In retrospect, however, when we first introduced the I -index, we should have cited Downhower et al. (1987) and Crow (1958) in order to emphasize that the I -index was not a term in our equations. We now make these citations on line 186, and this clears up why we define the I -index here, but then don't include it as a variable in Eqs. 3.1-3.3. We simply want to state that the I -index is a special case of the M index, but under empirically implausible assumptions about age-structure and exposure time to risk of RS . By relaxing these implausible assumptions, we derive the M index.

3. Some are not explained, e.g. SOM . Although most readers probably know what that means, I think it is much easier to understand if explained the first time it is used.

We included the text " SOM " only as a placeholder. Following acceptance, typesetters generally replace this text with hyperlink citations using the journal's own citation format for Supplementary Online Materials. To avoid confusion, in our revision we now use "supplementary appendix".

4. is there a reason why some of the equations are in the text, while others have their own line?

In mathematical writing it is standard practice to set-off larger or more significant equations by giving them their own lines, while integrating small or basic definitional equations into the flow of text. We have kept with standard conventions (see Higham's 2019 *Handbook of Writing for the Mathematical Sciences*, page 18).

Reviewer(s)' Comments to Author:

Referee: 1

The authors here propose a new way to measure skew (the M index) that they argue improves upon the B by addressing a key limitation. I am not yet convinced of this for three reasons:

1) It is unclear how M values are actually generated and therefore is rather opaque in its use as compared to the B.

In Eqs. 3.1 and 3.2, we provide a complete mathematical definition of M. Then, in Section 4, we provide a full derivation of M, and show some alternative representations. There is no opaqueness or ambiguity here—at all; M is fully mathematically specified. Moreover, in Eqs. 5.2 and 5.3, we show how our index can be converted to B using only basic algebra. M is not more complicated or opaque than B. It simply corrects a scaling issue with B that leads to serious inferential problems for comparative studies of reproductive skew.

Beyond the fact that M is fully defined, we also provide an R package that readers can use to generate M values from their own data. This R package is open source, publicly available, and linked in our manuscript on line 448.

2) I would argue that the ‘bug’ in the B index that M fixes, is actually a biologically meaningful feature and therefore M simply exchanges one bias for another.

While B is the most commonly used metric of skew, it is facile to show that it is strongly biased in ways that preclude comparisons across groups and species. To reiterate our point above, this statement is not based on personal opinion, it is based on mathematical proof. Nonacs’ B index changes structurally with sample/group size in ways that significantly impact evolutionary and biological inference. We provide a simple algebraic proof of this claim on lines 355-365. We conduct a simulation check that also establishes this claim in Figure 1. We also conduct a reanalysis of the best comparative database of reproductive skew and socio-ecological covariates available to establish this claim. We conduct both a direct reanalysis, and an improved reanalysis that uses more advanced statistical tools than the initial publication (Section 6).

We initially tried to use B for our own large-scale comparative study of reproductive skew, but it quickly became apparent that comparative study of skew with B could not be done. For example, samples taken from the same group of people (e.g., the Tsimane of Bolivia), with the same age-specific pattern of marriage and fertility showed radically different B values—a finding which was *clearly not indicative of biologically meaningful skew differences* between groups (given that the group itself and the structure of reproduction inside that group was held constant). We saw that this effect was driven entirely by sample size, which led us to discover the strong dependence of B on sample size that we prove algebraically in Section 5(a), lines 355-365, and via simulation in Figure 1.

Further, it is unclear to us what bias the reviewer believes to be inherent in our new M measure, as this is not stated explicitly. However, as we show via simulation and robustness checks on 270,000 simulated datasets (see Figure 1), we can demonstrate that M is *not* biased by sample size, mean RS, or age structure. This claim *cannot* be made for any of the other commonly used skew measures.

3) No existing index, including the B and apparently the M, address some real and fundamental problems in interpreting index values and particularly in across-population or across-species comparisons. Hence, it is unclear what the M index would add to solving such problems.

Again this is an assertion without evidence. Note, however, that we make no claim that M addresses all “real and fundamental” problems in comparing skew across populations or species. We simply show that one fundamental block to such comparisons is variation in group/sample size, mean RS, and age-structure. We then derive a skew metric with the correct scaling properties to deal with all three of these issues simultaneously—this is what M adds that no other metric does, including Nonacs’ B. M thus solves key problems in empirically modelling reproductive skew across populations.

First off, the B index is both mathematically and intuitively simple. I have made a program that researchers can download and use to calculate a B value from their datasets. However, they could easily make such calculations themselves. This makes it quite accessible to even mathematically-challenged experimentalists... Furthermore, unlike for B, there is no simple formula to calculate an M value. People will need to download a program from the authors (which is a good thing – thanks!). Personally, I wouldn't like relying on other people's black box programs to calculate an M value when it is not clear what it actually is summing.

It seems contradictory to propound the availability of a downloadable program for B, but criticize that for M. More substantively, our software is not a black box—all of our code that goes along with this manuscript has already been uploaded for review and public scrutiny to GitHub. It is open source and is publicly available for unrestricted, critical inspection. In this paper, we also fully specify the mathematical underpinnings of our code, and we conduct a rigorous set of robustness checks, as shown in Figures 1, 2, and 3.

With respect to the point that "...unlike for B, there is no simple formula to calculate an M value...", this is false, as we give a full mathematical description of both M and the simpler \hat{M} in the main text, Eqs. 3.1-3.3. A mathematically-challenged researcher can follow two simple equations, and a mathematically illiterate researcher can run 1 line of R code to calculate an M value. The code for the basic \hat{M} function is only 187 *characters* long, and hardly constitutes a "black box" incapable of review. Moreover, we show that our \hat{M} measure can be converted to B using only basic algebra in Eqs. 5.2. and 5.3. This is clearly described under the heading "Relation to other measures: (a) Nonacs' binomial index," lines 365-370.

In reading this manuscript, I found it very difficult to slog through the paper – it seems to be written more for mathematicians than biologists. Tactically speaking, if M is to replace B, you have to be able to explain it better than this!

The reviewer seems to making two distinct claims here: first that the paper is written for mathematicians, and second that the math, in general, is hard to understand, limiting the potential use and impact of our new skew metric. Regarding the first point, we note that this paper is written for biologists (and we try to make this clearer in the revised introduction, lines 13-74). However, for a mathematical measure to be used widely in biology, there is an important need to study its mathematical properties—otherwise we risk fostering a body of empirical research filled with years of scientifically incorrect inferences (as is the case with much empirical work based on the B index). Our paper, however, is not simply a mathematical exercise. We ground the motivation for derivation of our M metric in a problem that is central to the study of social evolution: we need to be able to quantify inequality/skew in fitness across populations in order to understand role of specific covariates on the evolution traits and dynamics that are of wide concern to biologists and other researchers.

In response to concern that the mathematical content of this paper will prevent people from using the M index, we emphasize here that to facilitate understanding and use of M by biologists and anthropologists without mathematical training, we have also taken several steps that go above and beyond simple presentation of in-text equations:

- 1) We carefully derive M, and provide a mathematical explanation of what it represents. (Section 4). Not every biologist will choose to carefully digest the derivation, but there is no alternative to advancing theory and method in quantitative biology than to formally derive and validate new measures.
- 2) We illustrate how M is related to other measures of skew and variation, so that researchers can easily convert between these measures (Section 5) and thus draw on larger bodies of literature for meta-analysis, without having to re-derive these relations themselves.
- 3) We provide an R package (SkewCalc) that allows biologists to calculate M values (and B values) with no mathematical training whatsoever.

- 4) We provide empirical data from our own field-sites in the R package, and provide a set of example vignettes (<https://github.com/ctross/SkewCalc>) that guide the reader through: i) data processing and calculation of skew metrics, ii) data simulation and robustness checks, iii) comparative analysis of skew by sex and cultural group, and iv) visualization of results and credible intervals on skew estimates.
- 5) We conduct an example cross-species comparison using phylogenetically controlled multi-level models, and include all of the code needed to reproduce this analysis and apply the same workflow to new data.

In sum, our contribution here extends beyond simple mathematical definitions (which we agree are necessary but not sufficient for our contribution to be both scientifically sound and of general interest to biologists). Instead, we provide mathematical derivations, wide ranging robustness checks of simulated data to validate the quality and scope of our metric, a software package to make implementation of our metric easy for non-mathematicians, and vignettes and statistical models to teach end-users how to use our software in research contexts.

Specific problems include the multiple variables and abbreviations (RS, for example, appears de novo in the body of the paper. Is it a variable? The output of a function? What? It wasn't until I went back to the Abstract that I realized it is just a general concept: Reproductive success!).

This is a fair point. We have now defined RS a second time—once in the abstract, and once on its first use in the body of the paper (line 51).

For clarity, we have also consistently offset variables and functions in their own typeface, and use only single letter symbols for variables and functions (in Section 5, however, there is a minor exception for the terms like “cv” and “var” which by convention appear as multi-letter symbols).

It would be very helpful to have everything listed in a table for quick reference.

We unfortunately do not have the space to repeat all of this text in a table. Instead, we have made sure that all symbols are properly defined on their first use.

There is also a use of terminology that does not relate to previous papers. For example, using the word “sample” when you really mean group size. Sample = the # of groups in your dataset since each group produces only one index value. Also, “population” (line 514). What do you mean here?

The reviewer claims our use of terminology does not relate to previous papers, but our terminology is consistent with statistical terminology going back at least 300 years. In statistics, a **sample** is a set of observations collected or selected from a **population** by a defined procedure (Peck, Olsen, and Devore, 2008; *Introduction to Statistics and Data Analysis*). Likewise, a **population** is the complete set of similar objects or events which is of interest for some question or experiment: a population is an arbitrary collection of elements, a sample just a subset of it (Bohm and Zech, 2010; *Introduction to Statistics and Data Analysis for Physicists*).

The general aim of statistical analysis is to produce an estimate of the properties of some chosen **population** (for example, the level of reproductive skew in Zebras in Bwindi Impenetrable National Park) on the basis of some **sample** (for example, the reproductive outcomes of 100 Zebras in Bwindi Impenetrable National Park that were randomly tagged and subjected to focal observation for a period of 12 months).

So contrary to the reviewer's assertion, when we use the word “sample” we do *not* really mean group size. **Group size** in the biological literature generally maps to **population size** in the statistical methods literature. The ratio of the size of **sample** to the size of the **population** is called a **sampling fraction**, and “group size” and “sample size” are only equivalent when the sampling fraction is 1; i.e., a complete census. While there are few species where complete census of reproductive outcomes per

group is feasible and even common, it is certainly not the case generally. Field observations are typically taken from a set of sample individuals that is smaller than the total group/population size.

In our simulation study, we define the reproduction parameters of the **population** using a standard statistical distribution for RS; i.e., the Negative Binomial. We can then simulate the RS outcomes of a **sample** of individuals from this population. This lets us show the effect of sample size on estimates of M, B, and other skew metrics, on average, as we hold the mean RS rate, over-dispersion level, and population age structure, constant.

In my opinion, a great advantage of the B index is that there are meaningful, diagnostic values. $B = 0$ means that reproduction is random. $B > 0$ means that reproduction is skewed (although maybe not significantly different from random), and $B < 0$ means that reproduction is shared more equally than a random expectation. It is not clear to me that M values will have such straightforward diagnostic properties.

In our paper, $M=0$ means that RS is random under a multinomial model with equal rates, $M>0$ means that reproduction is positively skewed (although maybe not significantly different from random), and $M<0$ means that reproduction is shared more equally than expected from a multinomial allocation with equal rates. These are the *exact same conditions* as claimed above for B, just derived under a more general framework. These conditions are apparent from Eqs. 3.1, and this is now described on lines 206-211 of the main text.

Second, the bug in the B is what is known as mean dependency. More productive groups produce different B values than less productive ones, even with identical skew functions. Consider 2 groups of 5 individuals and the number of offspring each produces:

A group from species A: 5,4,3,2, and 1

A group from species B: 500,400,300,200, and 100

Although proportionally, individuals in both groups are identical, the B index (when simulated) will reject Group B as being explainable by random processes, but Group A could be random. The difference is that distributing 1000 offspring gives a better measure of skew than distributing 10. Biologically, larger groups will tend to be more productive and therefore total number of offspring will tend to positively correlate with group size. Such larger & more productive groups mathematically allow a better estimate of their skew than do smaller & less productive groups. This is, in my opinion, is a positive feature of B, not one in need of fixing.

We are in perfect agreement concerning the point that “larger & more productive groups mathematically allow a better estimate of their skew than do smaller & less productive groups”. However, we note that following the laws of probability theory, and Bayes' theorem specially, this should be reflected in more *precise posterior estimates* of M as sample size or RS rate are increased (exactly as we show in Figure 3), *not* in a different expected value of the metric itself that depends explicitly on sample size! It seems that the reviewer tried to incorporate an assessment of confidence into the definition of the B metric using some ad-hoc method, rather than using the fundamental workhorse of probability theory, Bayes' rule, to derive posterior probability estimates of B conditional on a dataset. This upfront admission here appears to clarify where the sample size bias in B comes from. In contrast, we show how to correctly let the data impact the precision of measurements of M via Bayesian methods in Eqs. 4.8-4.9.

Next point: a thought experiment. Let's not think of groups A and B in the above example as different species. Instead, let's think of A and B as the exact same set of individuals. But, let's now let the numbers given above represent wealth in terms of dollars (case A) and cents (case B). The reviewer's assertion is that skew in wealth (this is the *expected value of the skew/inequality metric*) is different between cases A and B. However, we also know that nothing changed between cases A and B in terms of real wealth (only the units with which we choose to measure wealth in have changed), so the skew in wealth couldn't have changed either. Under the reviewer's example, an arbitrary decision as to how we define units now changes the expectation of the inequality metric. Simply stated, it should

not be the case that dollars are equally distributed while cents show a skewed distribution—this just isn't possible. The same issue would hold true if case A was a measure of biomass in terms of kilograms and case B was a measure of biomass in terms of grams. Useful metrics of inequality—be it in wealth, biomass, RS, fitness, or anything else—will generally be unitless (and thus have an expectation which is invariant to the units in which inputs are measured). Note from Eq. 3.2, that \hat{M} meets this criterion, as does the gold-standard measure of inequality in economics, the Gini coefficient.

Now, if we were to calculate posterior confidence in M on the basis of samples A and B above, we would get more precise estimates when the units are measured to the nearest cent/gram rather than nearest dollars/kg. With more precise measurement of inputs, we get more precise measurement of M . This relationship is shown in Figure 2(b), where we show that increasing mean RS rate yields tighter posterior estimates of M (without changing the expected value of the metric).

Our Bayesian method of estimating M formalizes the reviewer's intuition that "larger & more productive groups mathematically allow a better estimate of their skew than do smaller & less productive groups", but it does so using the direct application of universally accepted methods in probability theory to the problem, rather than by applying some ad-hoc, and in the case of B, mathematically biased procedure. The reviewer should also keep in mind that group size and sample size are not strict equivalents (as we described above), and B is structurally biased (in a formal sense) by sample size, not group size *per se*.

To the degree that groups A & B would have similar M values, I would view that as a flaw and not a benefit.

The reviewer's example above, however, misses the critical point we make in our paper. Our *critical* point is not that the B index is flawed because populations A and B above have different values (this mean dependency is a comparatively minor flaw in B). Rather, it is that we could pick *either* population A or B above, and just by taking the B index of two copies of the population (i.e., using 10 instead of 5 focal individuals but keeping the RS distribution constant) get a different value of the B index! *The most serious bug in B is not mean dependency, it is sample size dependency.*

It's a little clearer to see the effect if we use population C, where the vector (150, 50, 5, 5, 5) gives the RS count of each of 5 focal individuals. The B index according to Nonacs' formulation is 0.33. However, if we take a larger sample of focal individuals, *with the same RS distribution*, simply by *replicating* and *concatenating* the vector above 100 times, then the B index is 0.0033, a huge 100-fold change in value. *This* is the problem we take issue with (and since mean RS is fixed here, this behaviour isn't explained by mean dependency). The M index in expectation, however, is essentially invariant to such a change in sample size.

R code below (load SkewCalc and use ?M_index or ?B_index for more details about these functions):

```
> B_index(rep(c(150,50,5,5,5),1),rep(1,5))
[1] 0.3387345
> B_index(rep(c(150,50,5,5,5),100),rep(1,500))
[1] 0.003378135
> M_index(rep(c(150,50,5,5,5),1),rep(1,5))
[1] 1.693602
> M_index(rep(c(150,50,5,5,5),100),rep(1,500))
[1] 1.689145
```

This second flaw in the B index seems to violate the intuitions of the law of large numbers—i.e., that the expectation of our skew measure calculated on a sample should converge with greater accuracy to the "true" level of skew in the population as more data are collected and integrated. Instead, B simply converges to 0 for large samples regardless of the true level of inequality in reproduction in the

population. We demonstrate this algebraically in Section 5a (lines 355-365) and through simulation in Figure 1, which has been revised to include new subfigures on this point.

It is simply not a biologically meaningful feature that a skew measure changes structurally toward zero based on how many samples are used to estimate it. Imagine a population of tens of thousands of Zebras that have some level of reproductive skew that we wish to estimate. Imagine we study the population, and do focal observation of 50 zebras at random and count babies, then 150, then 250, then 500, then 1000, then all 10000. What we show in Figure 1 of our paper (and also algebraically in Section 5a), is that B will go to zero as we conduct more research *independent of the level of inequality in RS!* This is not biologically meaningful; this is a flaw in that skew metric. Simply stated, the theoretical contribution of any paper that uses B to compare between samples/groups of different sizes is highly likely to be compromised, as we demonstrate in our reanalysis of primate skew (Section 6).

M, in contrast to B, behaves properly according to standard statistical heuristics: the expected value remains constant as a greater number of samples are drawn from an RS distribution of fixed shape (Figure 1); and, the posterior uncertainty around this constant expectation decreases with increasing sample size or RS rate (Figure 2).

M accounts for sample size, mean RS rate, and age-structure. It does not trade off one bias for another in any test we have done. In sum, B has a very serious bias that M avoids.

The issue of mean dependency and group size was examined at length by K. Tsuji (Tsuji & Tsuji 1998 and Tsuji & Kasuya 2001) and he recommends the Morisita index as a way to avoid it. I am surprised the authors neither cite these papers, nor consider the I-indices in their analyses, as they explicitly already address the same issue this paper does.

The reviewer argues that we did not consider the I-indices in our paper. However, we explicitly showed how the I-index relates to \hat{M} in Eq. 5.4, and on lines 186 and 262-265. We had already proven in the first draft of this paper that the I-index is just a special case of \hat{M} under unrealistic assumptions about age-structure. In Eq. 4.2, we derive \hat{M} as a mathematical generalization of the I-index that accounts for the effects of variable exposure time to variance in reproductive success. Note, for example, Eq. 4.2h: if $\text{corr}(r, R \hat{t})^2$ goes to zero [e.g., if \hat{t} is fixed at $1/N$ for all i], then we recover the I-index exactly. We also included the I-index and the I-index of RS rate in Figure 1 in our comparison across metrics. Indeed, the I-index took up 40% of Figure 1.

The Morisita index, I_σ , is basically just the I index with a sampling correction (i.e., the I_σ index is to the M index what the I-index is to the \hat{M} index). As the reviewer has argued in published work, it is important it deal with variable exposure time to risk of RS (this is after all what B was supposed to adjust for), so neither the I-index nor the I_σ index are general solutions that will permit comparative work. However, since I_σ is so similar to the standard I-index, we now include it in Section 5, lines 380-390.

Also, we now cite the suggested papers on lines 60-65 and 380-390, and explain why they don't solve the issue at hand. The reviewer is correct, however, that these papers should have been cited in the original draft. This was clearly an oversight on our part.

Third, if you have only one group (like either A or B from above), all skew indices are very limited in the conclusions one can draw. For B, yes one can reject a random distribution, but the observed distribution could result from any number of underlying skew functions. For example, if one group from species C had a distribution of 550,425,300,175, and 50; it would not be possible to conclude that species B and C differ in skew although they would have different B and M values.

This assertion is not supported by evidence. Moreover, it is an assertion that directly conflicts with Bayes' theorem as we describe below, and, in fact, already described in detail in the submitted version of our paper (in the caption to Figure 2).

The assertion that one can make a comparison with “a random distribution” but no other distribution is simply incorrect (see a recent textbook critique of this logical fallacy below). While the reviewer is correct that such tests were *at one time* the most computationally feasible ones to conduct, this is not true given the computational resources we have available in 2020. One can simply calculate the posterior distribution of M applied to the RS data from populations B and then C, and then evaluate the distribution of the *difference* of these M values. One can then check the overlap of this distribution with zero to evaluate the statistical reliability of a difference: it could be that the distribution is reliably non-zero, or it could be as the reviewer claims above. This, however, is an empirical question to be addressed by estimating the posterior distribution of a difference, not by asserting the null. We use textbook methods in our paper (see below from Richard McElreath's *Statistical Rethinking* [2020], page 158):

“If you want to know the distribution of a difference, then you must compute that difference, a contrast. It isn't enough to just observe, for example, that a slope among males overlaps a lot with zero while the same slope among females is reliably above zero. You must compute the posterior distribution of the difference in slope between males and females. For example, suppose you have posterior distributions for two parameters, β_f and β_m . β_f 's mean and standard deviation is 0.15 ± 0.02 , and β_m 's is 0.02 ± 0.10 . So while β_f is reliably different from zero (“significant”) and β_m is not, the difference between the two (assuming they are uncorrelated) is $(0.15 - 0.02) \pm \sqrt{0.02^2 + 0.12} \approx 0.13 \pm 0.10$. The distribution of the difference overlaps a lot with zero. In other words, you can be confident that β_f is far from zero, but you cannot be sure that the difference between β_f and β_m is far from zero... Lurking underneath this example is a more fundamental mistake in interpreting statistical significance: The mistake of accepting the null hypothesis. Whenever an article or book says something like “we found no difference” or “no effect,” this usually means that some parameter was not significantly different from zero, and so the authors adopted zero as the estimate. This is both illogical and extremely common.”

Using the data in the reviewer's specific example, applying Bayes' rule to estimate M, and following the above statistical guidance for estimating posterior contrasts, there is actually a small but reliable difference in M, 0.12 (90%CI: 0.04, 0.20), between distributions B and C given by the reviewer.

See code below to replicate:

```
library(rethinking)
library(SkewCalc)

M_index_stan(r=c(500,400,300,200, 100),t=rep(1,5))
M_b <- extract(StanResults, pars="M")$M
M_index_stan(r=c(550,425,300,175, 50),t=rep(1,5))
M_c <- extract(StanResults, pars="M")$M

mean(c(M_c-M_b))
HPDI(c(M_c-M_b),prob=0.9)
```

In the caption to Figure 2(a), we walk the readers through this exact type of comparison, but where the parameters above are instead posterior estimates of M from different cultural groups. We specifically show that reproductive skew between polygynous Kipsigis men is reliably higher, as distinguished by Bayesian posterior estimates of M, than reproductive skew in either the serially monogamous Afrocolombians or the monogamous Embera. We also show that reproductive skew in the Afrocolombians cannot be reliably distinguished with high probability, via Bayesian posterior estimates of M, from reproductive skew in the Embera—even though point estimates of skew differ

between all three groups. We do not, however, assert the null, like the reviewer does above; we present the credible interval of the contrast itself.

The reviewer is both wrong about the details of how to conduct this statistical procedure, and writes as if we were unaware of the fact that point estimates don't illustrate posterior credible intervals, even though we explicitly raised this issue in our paper (Section 4b) as a motivation for avoiding the use of point estimates outright and instead using more contemporary Bayesian estimation methods that fully resolve such issues. We also provide software that automatically does such computations, and we provide vignettes that walk end-users through conducting such tests in a step-by-step fashion (see readme file on SkewCalc Github page, or the example code listed just above).

And basically if all you have is A, there is literally nothing you can conclude about species level skew.

This assertion again violates the laws of probability theory. Following Bayes' theorem, we know that in principle *any* data allows us to update our understanding about a parameter. A small amount of data provides very little information, a large amount of data provides a lot of information. If you have only the data given in A above, you *can* say—for example—that skew at the species level is of perfect equidistribution with probability zero (and this is more than one could say without the data provided in A). We can also run the following code:

```
M_index_stan(r=c(5,4,3,2,1),t=rep(1,5))  
M_a <- extract(StanResults, pars="M")$M
```

and find that the data in A is consistent with population levels of M ranging -0.40 up to 0.66 (these are bounds on the 95 percent credible interval). This is, of course, an enormous range, so from A alone, we cannot make any strong claims—but we can say more than “literally nothing.”

What is true is that use of *point estimates of B or M* would be misleading when taken from small samples, like A, as they would overstate the weight of evidence in favour of a particular skew value given the amount of data. This is why we raised this very issue in our paper and derived a method of estimating the posterior distribution of M in Section 4b (and provide an R package that does this automatically for other researchers, as we show above), and why we are strong advocates against use of point estimates whenever possible. See for example our comments on lines 298-300, 312-335, and in the caption to Figure 2.

What this means is that comparative studies are at the mercy of sample sizes (i.e., the number of groups in your dataset), the size of those groups, and the number of offspring produced per group. This means you need to calculate B or M values for multiple groups and then simulate whether the average index value could have been produced by chance.

We understand the intuition the reviewer has here, but the bootstrapping algorithm described above is simply an approximation for doing full Bayesian inference and meta-analysis (see for example Ruben 1981). Before we had fast computers, null-hypothesis rejection approaches were the only thing that was feasible. This is no longer true, and we already do better using the Bayesian approach in Section 4(b).

The reviewer here is critiquing us for not being aware of something that we are keenly aware of. Indeed, as just noted above, our Bayesian approach in Section 4b exactly addresses what the reviewer proposes here, but uses more contemporary Monte Carlo estimation methods that correct some problematic assumptions of standard boot-strap resampling (e.g., the fact that unobserved sample outcomes have zero density in the standard bootstrapping approach, even if they may have non-zero posterior probability in a biologically reasonable generative model).

For example, if a dataset consists of 10 groups just like A, then the B index would find that no

individual group is significantly skewed, but that the population is. This sort of simulation is essential for almost any index being used to determine a species-level character. (Note this is what my B index program does and why people might want to use it.)

First point: the idea that we should only be interested if any particular group has a skew value that is significantly different from random is over-simplistic and precludes more nuanced tests of theoretical models. Instead, it is now common to use full Bayesian inference and formal meta-analysis methods to perform the kinds of estimations described above (See McElreath's *Statistical Rethinking*, 2020).

Second point: this is why our program outputs full Bayesian posterior estimates and not simply point estimates. Our model already does what the reviewer asks for, but under a more rigorous estimation strategy: indeed standard bootstrapping algorithms tend to overestimate confidence in skew metrics from small data sets by assuming that the population distribution is of the exact same shape as the sample distribution. The Bayesian approach that we use relaxes this problematic assumption.

The inferential goal is to estimate the posterior density of our M index conditional on a data set. It is clear that the width of the posterior distribution should shrink as more sample individuals are taken from a population of fixed size and used to estimate an index, be it M or B; our model behaves correctly in this respect as shown in Figure 2(b). Estimates of M from small samples have huge confidence intervals, because there is not enough data to distinguish between possible generative models. If one has a large set of data from many groups with small samples per group, then one can use formal meta-analysis methods on the posterior estimates given by our software.

Finally, the reviewer's package calculates B, which we prove is biased by sample size. Averaging a number of biased estimates does not remove bias no matter how many replicates are done (we show this via simulation in Figure 1), so the package that the reviewer provides does not resolve the issue at hand (which is providing estimates of skew that are unbiased by group/sample size).

Thus, the authors' comparisons of individual B values (Figure 1) is meaningless because individual B values are meaningless. You need to simulate confidence intervals.

Yes, the reviewer is right that confidence intervals (or better, Bayesian posterior credible intervals) are needed around point estimates of either M or B. As did the reviewer, we also provide software that does this. Note, however, that we use Bayesian methods to estimate the probability density of M conditional on a data set. The reviewer's approach, in contrast, can only rule out the probability of a B value under a null model, whereas our methods also allow more nuanced contrasts between non-null models (see comments above and our detailed empirical example in Figure 2a).

Additionally, the reviewer is factually mistaken in his wider claims here. First, Figure 1 does not show individual B values, it shows the mean index values with shaded 95% confidence intervals over 500 simulated data sets for each value of sample size on the X axis. The issue is that for MMP, λ , and B, the scale of the bias introduced by sample size dependence is so large that the scale of the confidence intervals on the mean can't be seen when plotted on the same axes!

We, however, did not describe Figure 1 sufficiently in the first submission, and the lack of clarity here is squarely our fault. To make it more clear that we do exactly what the reviewer suggests, we have added two new insets to Figure 1, and updated our caption to state that the number of replicates that were done per value of sample size is 500 and that confidence intervals were also plotted, even though they can't be seen.

This brings me to the authors' reanalyses of the Kutsukake & Nunn paper. I haven't read that one in a while, but one problem of great effect may be the use of species means. As I argue above, this might mean that the data are far more definitive for some species (i.e., more groups measured, larger, more productive) than for others (i.e., fewer groups measured, smaller, less productive). Thus, all species means are not created equal and to treat them as such in any

analyses can give biased results.

We used species means only in our most basic direct replication of the study by Kutsukake & Nunn. We offered this same critique of the use of species means ourselves. This was one of the reasons we listed on lines 420-428 for including Section 6d—a phylogenetically controlled, multi-level Bayesian analysis of group-level, not species-level, data—in our original submission.

In our main models (Section 6d), we do not use simple species means. We use phylogenetically controlled, multi-level Bayesian models that were designed specially to deal with unbalanced sample sizes (of $n=84$ groups in this case) by species. This was described in our Supplemental Materials in full mathematical detail, and in Section 6d of our submitted manuscript.

What I would suggest is that confidence intervals for each species be derived, with the analyses consisting of a 1000 or more random draws from these distributions. I don't remember if K&N did this and I don't think the authors here do it. But they should.

We strongly agree with the reviewer that comparative analysis of skew should be done using full Bayesian posterior inference, rather than point estimates. This is why we have reached out to more than 70 colleagues across the evolutionary sciences to obtain the individual-level data needed to make such posterior estimates of M in 97 small-scale human populations and 76 species of non-human mammals: our empirical paper on this topic is under revision.

However, prior to our empirical work, the best open-access available data for comparative study of skew was provided by K&N. This data set provided by K&N does not include the individual-level observations needed to estimate posterior distributions (or even conduct bootstrap resampling). So it is simply not possible to do in this paper.

Keep in mind that our point in re-evaluating the K&N results with M was not to conduct the most rigorous comparative analysis of skew possible (we attempt to provide *that* in our upcoming empirical paper using our own data). Instead, our point was simply to show that the results presented in K&N and similar studies are fundamentally confounded by sample/group size, and that application of our skew metric (along with more advanced phylogenetic methods) can yield more insight into the relationship between skew and covariate data (this is described in Figure 3 and its caption).

The point the reviewer raises here was already mentioned in our original submission when we said:

“Our statistical work-flow would be improved by estimating posterior distributions for M using individual-level observations and then modelling this uncertainty in M as measurement error in the comparative analysis” on lines 431-435.

However, for us to challenge inferences drawn from the best-available data-set in the literature, due to their reliance on faulty metrics of skew, we have to reanalyse that same data-set using new tools and present the inferential contrasts. Holding all else constant, we show that using a biased metric to estimate skew leads to incorrect inferences about what is associated with reproductive skew. K&N mistakenly infer that the covariates associated with the bias of the measurement tool are actually associated with reproductive inequality/skew, when this is not the case. Our analysis allows us to show this, even if the data-set provided by K&N is not as fine-grained as we would hope for.

Finally, the above simulation would require fitting a best fit function of skew to each dataset. That, of course, assumes that one such function exists and variations across groups in skew is due to stochastic processes around that function. That, too, is an assumption – different groups within the same species or population may have radically different skew functions. For example according to skew theory, groups of close relatives ought to divide reproduction differently than groups of distant or unrelated individuals. Therefore, if this is the case, and one does not know the kin structure of the groups in your dataset, trying to fit one function to the entire dataset will be misleading. Again, this

may be problematical with the K & N data if there are intrinsic reasons why skew would vary across groups within a species.

Addressing how various covariate features of a population (e.g., relatedness) might influence skew seems quite tangential. The author is right that theory identifies relatedness as an important predictor of skew, but we do not see this as germane to the issues under discussion in our paper—it is something to instead take up with the original work of K&N.

Our goal in the analysis was not to show that there exists “one best fitting skew function”. Our point was to show that the most recently published, and highest profile comparative study of reproductive skew reached incorrect conclusions because it used metrics that were inherently biased. Direct replication of the original study required us to use methods consistent with the original paper. However, we didn’t just do the minimum. We also applied cutting edge statistical models to the original data and conducted a second reanalysis, which simply confirmed the findings of our initial reanalysis using more basic methods.

Finally, the reviewer’s point about investigating the role of some variable (e.g., kin structure) is actually what motivated us to derive M in the first place. We want to be able to conduct studies analyzing the effect of various factors on reproductive skew in comparative context. The problem, which we have established using basic algebra, simulation, and replication of extant empirical studies, is that this cannot be done with current skew metrics like B. The B index is highly biased by sample size dependence, and this bias will lead to confounding in comparative studies of skew where there is variable group/sample sizes; hence a new metric is needed.

Referee: 2

Comments to the Author(s)

Dear authors,

A new measure of reproductive skew which controls for differences in group size between samples is a valuable contribution to the field of social evolution and behavioural ecology, which is why I find the manuscript of sufficient general interest.

We thank the reviewer for this positive evaluation and agree that a measure of reproductive skew which allows for comparisons between populations/species of different sample/group sizes is essential for comparative analyses in the fields of social evolution and behavioural ecology. We have aimed to introduce a new metric of skew that will not only be as easy to implement as the highly impactful B index introduced by Nonacs (sources 19 and 20 in our paper have more than 250 cumulative citations), but will also allow for meaningful comparisons of skew *between* populations. This is an emerging area of research, but one that has been held back by the lack of a usable, unbiased metric for measuring skew.

Though I suggest the following changes before publication.

The introduction of the manuscript and the introduction of the new index is well written and comprehensible.

To address the editor’s emphasis that we spell out the “potential impact, through enhancing in an explicit manner, how this work significantly advances the field, with an increased likelihood for uptake and more wide-scale applicability”, we have tried to more explicitly comment on why M is needed to advance the literature on the drivers of reproductive skew in comparative context. However, in light of the comment above, we have also tried to avoid disrupting the flow of the introduction that Reviewer 2 finds so comprehensible and well-written.

Specifically, in Section 1, lines 75-115, and Section 2, lines 118-174, we now detail why comparative studies of skew are so important, and why existing metrics have failed to permit such tests; but otherwise, we have kept consistent with our original prose.

In the introduction of the new index (section 2. A comparable measure of skew, line 105 – 148), “X” and “E” in equation 2.1 as well as the equation 2.3 is not further explained.

The “ \square ” symbol is a standard mathematical symbol that is unambiguously understood to mean “expected value” across all fields of quantitative science. Nevertheless, we indicate what it means on line 182 when we say that:

“Eq. 3.1 defines $M(r,t)$ to be the difference of the observed estimate of $\hat{M}(r,t)$ **from the expected value of $\hat{M}(r,t)$ if RS were distributed as a multinomial outcome with same total sample size, RS rate, and exposure time vector.**”

The bold section in the above quote explains exactly what Eq. 3.1 is doing, and what the symbol “ $\square[\hat{M}(X,t)]$ ” means.

All terms in Eq. 3.1 are fully defined. First, the \hat{M} function for two inputs is defined in Eq. 3.2. Then, in Eq. 3.3, we define X to be a variable with a multinomial distribution with size R and probability vector t . In other words, Eq. 3.3 is included specifically to define the variable X . This provides a complete definition of every term in the model.

In section 4 (Derivation) a long list of equations is listed which are not further explained and hence it is not clear of what use they are to understand the manuscript. This needs to be further explained/mentioned or I suggest excluding some of the equations.

This section is titled “Derivation”, and the series of equations in Section 4 is a derivation. These equations show the steps in the derivation of \hat{M} as a standardized conditional variance. We can’t just make the assertion without showing the derivation stepwise. Not every reader needs to use each equation here, but any reader should be able to follow the steps of our derivation of \hat{M} should they choose to in order to check and validate our work. Accordingly, we have decided to keep the equations in, and simply explain to the reader that these equations are a step-wise derivation. See lines 247-250.

In section 4, the new reproductive skew index gets set into relation to other skew measures. Here, I do not see what extra information is added by comparing the multinomial index, \hat{M} , to more than the Nonacs’ binomial index. Especially because the other skew measures (Coefficient of variation, opportunity for Selection and Gini coefficient) are not further mentioned in the manuscript. This needs to be further explained/mentioned or I suggest excluding all skew measures besides the B index for comparison with the multinomial index.

We have found that the empirical literature is filled with reports of skew under many different metrics. Comparative studies will need to put these metrics into a standard form. By showing how each metric relates to \hat{M} , we give biologists the tools they need to easily convert extant metrics into \hat{M} , and thus facilitate comparative work. We make this point explicitly now on lines 100-106.

In section 5, data from Kutsukake and Nunn (2006) is re-evaluated with the multinomial index. Here another set of skew measures used in Kutsukake and Nunn (2006) is compared with the new index. It might be useful to compare these skew measures (λ and maximum mating proportion) with the multinomial index in section 4.

This is a good idea, and one we initially tried to work through. However, both λ and MMP are fundamentally different measures of skew that have no direct algebraic relationship to \hat{M} . The best we could do is show how λ and MMP scale with sample size, mean RS, and age structure when applied to the same simulated data as \hat{M} and B (Figure 1).

In section 5a, b and c, information on the univariate associations between the skew measures, multiple regression analysis and on the step-wise multiple regression models are missing. Information on included predictors in models, used R packages, checked assumptions, and used statistical test should be at least stated in the supplementary online material and mentioned in the main manuscript text.

This information appears on lines 429-449. Additionally, the entire R script is included in the supplementary materials.

In the summary (section 5e) the authors state that the multinomial index and B index are highly correlated due to the small sample sizes in the Kutsukake and Nunn (2006) data set and expect larger differences between indices with data of larger sample size differences. Sample sizes of $N=2$ to $N=10$ are not sufficient for statistical analysis these days and I suggest comparing it to published reproductive skew data with larger sample sizes (e.g. Stier et al. 2011, Engelhardt et al. 2017, Surbeck et al. 2017, Minkner et al. 2018).

Among animals groups with small numbers of members, sample sizes per group will be small. Nevertheless, a skew metric can still describe how reproduction in those small groups is partitioned. This is a somewhat different issue than doing statistics on small samples. The N for number of observations in our statistical model in Section 6d is 84 groups. We have, however, cited the indicated sources, and integrated the discussion suggested by the reviewer into our paper. See lines 50-59.

Some general notes on the manuscript:

1. I am missing an ethical statement concerning the conducted human reproductive success research.

We have now added this to the Supplementary Materials, section 2.

2. The use of two terms (reproductive skew and reproductive inequality) of similar meaning is confusing and I suggest using one of the terms throughout the manuscript.

As described previously, we use both terms, but have been more clear about using “inequality/skew” when needed. See also footnote 1 in the main text.

3. Please check that you use introduced abbreviations all the time, e.g. MMP (line 404) as well as spell out SOM at least ones.

MMP, “maximum mating proportion”, is introduced, defined, and cited on line 404. We now use the acronym, MMP, instead of the complete phrase, after first use.

4. Concerning data availability, I suggest making more clear where to find the human populations reproductive success data (Kipsigis, Afrocolombians and Emberá) on the main authors GitHub.

The GitHub readme file linked in the paper (<https://github.com/ctross/SkewCalc>) now has detailed vignettes on how to access and analyse these data-sets using the SkewCalc R package.

Otherwise I enjoyed reading the manuscript and find it of high future value for comparisons between groups, populations and species research.

Many thanks for the constructive feedback!

Appendix B

Responses to reviewers:

Associate Editor:

We thank the authors for addressing some issues, and while there remain some substantive aspects for you to consider, we provide a final opportunity for you to revise your manuscript. Particularly the third referee provided very detailed comments, which will definitely help to further improve the manuscript, e.g. by clarifying if and how sampling at different life stages can influence the measurement.

We are grateful for the opportunity to provide a final revision. We have carefully read the reviewers' feedback, and responded to it pointwise below. We have integrated the reviewers' suggestions whenever possible. For example, we have clarified issues around sampling design and added a section to the supplementary appendix which generalizes our age-specific fertility adjustment to account for arbitrary functional forms linking mean RS and age. We also now reinforce our prior point that sampling and metric selection are issues with any study design, using any index, not an issue with the multinomial index itself. We do, however, cite Waples on this topic, as he offers a great review of how sampling can impact estimates.

We have also been explicit that Morisita's (1962) I_{σ} index to correct the I-index's dependence on mean RS (to adjust for sampling) was recently re-introduced by Waples (2020), under the new name of Δ_I . We do not consider this an issue for the novelty or impact of our paper, as our goal in developing M was not to correct the I-index's dependence on mean RS (a previously-solved problem), but rather to derive a generalization of the I-index that is robust to variance in age/exposure time. A first step toward this goal was introduced by Nonacs in the early 2000's, but sample size dependence renders his metric unsuitable for comparative studies of skew. In the last 20 years, no index has been derived that corrects the sample size dependence in Nonac's B, while maintaining its control for exposure time. M does, and this is novel and important for comparative research, even for comparisons of different populations of the same species, or the same population at different times.

Waples' index is essentially identical to Morisita's index up to an additive constant ($\Delta_I \approx I_{\sigma} - 1$)—see Section 5 and Fig. 1. I_{σ} was previously derived to achieve the same goal that Waples cites. Our approach, however, follows a mathematically distinct derivation, and was designed to generalize both the I-index (our \hat{M} -index) and its sampling corrected version, Morisita's I_{σ} index (our M-index), to account for heterogeneity in age/exposure time. M is a substantial generalization of the I-index, that goes well beyond what Waples has re-introduced. We aimed to derive an index that is robust to sample size, mean RS, *and* age structure: the metric described by Waples/Morisita does not meet these three requirements, and neither does Nonacs' B. M is still needed for comparative studies of skew in systems like humans and other mammals where age structure cannot be easily ignored. We do not claim that age/exposure time *must* be controlled for in all cases or study systems, but we argue that a metric which *can* account for its effects is an important contribution to the field.

As a brief overview of our changes, we make substantial additions to address Waples' concerns in sections 3(b) and 5(e) of the main text. To make space for these additions, we move a small amount of technical details on Bayesian inference from section 4(b) to the appendix section 3. We also provide a more detailed derivation of M as the first section of the appendix, and add a substantial generalization of M to account for highly-non-linear age-specific fertility functions to section 2. All other changes are generally typographical.

Referee: 3

In this paper authors develop a multinomial index of reproductive skew the main advantage of which is that it is not biased by differences in sample/group size, a problem with some existing indices, notably, the authors mention Nonacs Binomial Index B . The paper is well written and the subject is of relative importance.

We appreciate this feedback.

One concern lies in the degree to which this index, its assumptions and utility relates to an index of resource monopolization published over two decades ago, referred to as index Q and based on assumptions of multinomial distributions (Behavioral Ecology 1996 Vol. 7 No. 2: 199-207). How does the index of RS described in the present paper relate to the index Q introduced in that earlier study?

We thank the reviewer for flagging this article as a possible concern. M is structurally very different from Q. Q is, however, very similar to Nonacs' B. In fact, they both depend on sample size (i.e., they scale like $1/N$, where N is the number of individuals in the group/sample). In other words, it has the same key flaw as Nonacs' B, despite relying on the multinomial distribution tangentially in its derivation.

We now cite the paper, include the Q index in Figure 1, and show how it is mathematically related to M in Section 5(c), lines 408-411. We note that Q is not a useful measure of skew in comparative studies (when sample sizes differ) for the same reason we gave for B.

Referee: 2

I appreciate that the suggested changes and concerns for this manuscript were addressed and the manuscript was revised where feasible.

We appreciate this feedback.

The manuscript has improved and only minor changes are suggested. I would advise to further improve the manuscript by stating at first mentioning of the indices (multinomial index, binomial index, opportunity for selection, 'uncorrected' multinomial index) the term which will be used throughout the manuscript. As for example, throughout the manuscript up to five different terms are used for the multinomial index (e.g. L87: multinomial index, M, L88: M, L113: multinomial index, L194: M index,) and the binomial index (e.g. L65: Nonacs' B index, L85: B, L193: binomial index, B, L338: Nonacs' B, L346: Nonacs' binomial index). It would make the manuscript easier to read.

Thank you for this suggestion. In the interest of readability—and also word count—we have made our usage more consistent, sticking to single letter labels after first introduction.

Additionally, the use of the two terms skew and inequality were explained by the authors, which I appreciated. Though I suggest not using the two terms within the same sentence (L138-144).

We have revised this line to avoid the use of the term "reproductive inequality".

Last but not least, please replace 'reproductive success' with RS in line 174.

We have revised to 'an RS measure' instead of 'reproductive success', as it seems awkward to use an acronym (RS) right next to the variable name (r).

Referee: 4

This is a revised version of a manuscript that I did not review in the first round. I looked over the response letter but am evaluating the revised manuscript on its merits rather than trying to assess the adequacy of the authors' arguments. I think the manuscript deals with a consequential issue and could potentially be an important contribution. I also think it could be improved in a number of ways before publication.

We thank the reviewer for taking the time to give our paper a thorough read, and for providing such detailed feedback. We have revised the paper in several places in light of the suggestions below. Mainly, this involved being more careful in clarifying the conditions under which M may be usefully applied (see lines 214-264), and adding a subsection on how Waples' Δ_1 index relates to M (lines 420-428). We respond to each issue pointwise below.

The main issues involve the following:

- Clarifying differences between population genetic and social sciences perspectives on reproductive skew.

As we will argue below, this point is a good one. It does not conflict at all with our thinking on skew, but corrects a lack a nuance in our original writing. We should have been more clear about how failing to control for age-structure can be a problem in social science data (that we ourselves have used), but that in other systems, control for exposure time is not always necessary, or even desired. We have revised the problematic passages in this revision, as we describe in more detail below.

- Clarifying issues related to Crow's I

We provide a clarifying statement below, but note that our text was correct as written.

- Clarifying restrictive assumptions

We provide these new details as needed. We note to the editor here that our M index is actually a formal generalization of Waples' Δ_1 (which is itself equivalent to Morisita's I_σ index). Not only did we submit our work to peer-review first, but we provided a substantially more general solution to the problem of measuring reproductive skew in a way that permits comparative research—dealing with both variation in mean RS and variation in age/exposure time. In other words, if one wishes to control for age structure, then a vector of age/exposure times can be passed into M , but if one wishes only to adjust for mean RS, then exposure time can be set as a constant, and M will behave exactly like Δ_1 or Morisita's index. M is thus more broadly applicable for comparative studies than either Δ_1 or Morisita's index, and is a theoretical contribution beyond what is offered by those measures: as we have stated before, they are special cases of M , under assumptions about age-structure that do not always—*but might sometimes*—hold.

Waples' Δ_1 , being a special case of M , has, strictly speaking, *more restrictive assumptions* than M .

Waples' arguments all have to do with which sub-sets of data should be fed into M when trying to understand skew in a given population (e.g., should all living individuals of any cohort in a given year be used to estimate skew? Should all living individuals of the same cohort be used? Should all individuals ever born to a given cohort be used?). Each sub-set of data tests a different question about how reproductive inequality is parsed among different sub-sets of individuals. We make no claims about which data subsets should be used, because such claims cannot be made without reference to a specific research question. Furthermore, such questions are orthogonal to our goal of

providing a generalized measure of skew free of the biases inherent in previously-published indices. More details are given pointwise below.

- Life stage sampling

Again, this has to do with study design, and the methods needed to address specific questions, it is not a critique of the validity of M ; it simply forces one to think about which data should be fed into M , and if exposure time control should be employed or not. We comment on this issue on lines 214-264.

Regarding the first point, my interest in this issue comes from the population genetics perspective. In particular, the ratio of the variance in offspring number (V_k) to the mean (k_{bar}) largely determines effective population size and the key ratio N_e/N . For iteroparous species, which are the focus of this manuscript, N_e per generation depends heavily on lifetime V_k^* , which is measured across all individuals born into the same cohort (see Hill 1972, 1979). Two major factors that contribute to V_k^* are: 1) variance in seasonal reproductive success among individuals of the same age and sex; and 2) variance in adult lifespan, which determines how many seasons/years an individual gets to reproduce in. It can be useful to partition lifetime V_k^* into these two components, and that indeed has been an interest of mine, but in terms of N_e per generation what matters is overall V_k^* arising from both factors (as well as other factors mentioned below). In contrast, the authors appear to be looking at reproductive skew from a social sciences perspective, in which case it seems to be important to “control for the effects of heterogeneity in age or ‘exposure time’ to risk of reproduction.” I don’t have a problem with that, but I think it is misleading to dismiss as deficient some skew indices that don’t make this adjustment. For example, if you removed from lifetime V_k^* the component related to variance in longevity, you would remove a substantial component of the true signal that causes N_e to be generally less than N . On this point, here is a statement about the Opportunity for Selection from the authors’ response letter: “We simply want to state that the I -index is a special case of the M index, but under empirically implausible assumptions about age-structure and exposure time to risk of RS .” I don’t know what “empirically implausible assumptions about age structure” the authors are talking about; Crow’s 1958 does not explicitly deal with age structure. A more equitable presentation would recognize these inherent differences in perspective. This would not diminish the value of M , which the authors show is quite general and could be easily used either with or without adjusting for exposure time. If they simply are referring to the fact that some individuals, by chance, live longer and accumulate more lifetime RS , then I disagree that this is a problem for Crow’s I . If they are referring to something else related to age structure, they need to clearly explain what they perceive the problem to be.

This is a very good point. We agree with the reviewer entirely: a more equitable presentation would recognize these inherent differences in perspective, and we have revised this passage accordingly. We wrote the above quoted lines with a specific confounding issue in mind: in social science data, focused on a species with a slow life history and non-discrete, overlapping generations, we sometimes want a measure of variance in RS that includes all living individuals in a given population, but variation in age structure leads to issues in measuring variance in RS because some individuals have had more time to acquire RS . In some cases, we can sidestep the issue and collect data only from reproductively complete individuals (e.g., those over 60 years of age), but this comes with trade-offs in terms of power and the time-frames to which we can speak. Simply ignoring differences in age when using full population data is often problematic, because exposure time is variable.

The reviewer, however, is right that this does not mean that a metric which controls for exposure time is always needed: changing the demographic composition of the sample (i.e., selecting individuals of a specific cohort) is another option. The latter approach conditions on age via the

sample selection process, rather than statistically via use of a standardized age-conditional variance. Either option can be useful, depending on context. We have now changed the language so as not to appear to dismiss indices that don't adjust for exposure time. Those indices can be very useful for many research questions (e.g. population genetic analysis of lifetime RS).

On lines 41-45, we include the qualification: "Though the details of a given research question may sometimes necessitate other kinds of measures" before stating that: "researchers across fields as diverse as biology, anthropology, and economics generally agree on the following desiderata for comparative measures of skew/inequality: ...". We believe that the desiderata are generally sought-after in a skew measure, but we acknowledge that the details of a specific problem might justify other measures.

Likewise, on lines 214-216 we rehash this point: "Despite its comparative robustness relative to existing measures, M , is not universally applicable to all questions about reproductive skew; specific skew indices should be carefully chosen with respect to the scientific questions being addressed [34,48]." Lines 217-264 then provide an overview of many of the reviewer's points.

Discussion of Crow's I is inconsistent and sometimes contradictory. Crow defined I in his 1958 paper as the Index of Total Selection. More recent authors, starting with Wade 1979 Am Nat, generally use the term Opportunity for Selection, which seems more appropriate. Crow's 1958 paper was reprinted in 1989. The authors cite both of these papers separately (#40 and #69), which is confusing. In that 1989 volume Crow published a short update to the 1958 paper (Human Biology 61(5/6), pp.776-780), which contains some discussion the authors might find useful.

This double citation was a mistake, we should have used only a single citation. We have now fixed this issue.

Many authors have pointed out that Crow's I is negatively correlated with mean offspring number in the sample ($kbar$), which has limited its usefulness, esp when comparing estimates across species, studies, or even sexes. This dependence on $kbar$ is the major factor distinguishing the authors' proposed M from the binomial index B . In the text below Eq 4.2i the authors correctly note that this same dependency on sample $kbar$ applies to Crow's I ; however, earlier they say the opposite: "For example, the opportunity for selection index, I , and its sampling adjusted counterpart Morisita's index [46, 47], are unitless, invariant to sample size, and related analytically to variance in RS."

The mean number of offspring per individual in the sample (i.e., mean RS) is not the same as the number of individuals in the sample (i.e., N). Both of our statements above are correct and do not contradict each other. As we show, Crow's I -index is negatively correlated with mean RS (see Figure 1), but it does not depend on the sample size of individuals (again see Figure 1). Morisita's index corrects the scaling of Crow's I -index with mean RS. It does not account for variability in age/exposure time to risk of RS.

For clarity, we have nevertheless revised the quoted sentence above (lines 61-66) to state: "For example, the opportunity for selection index, I , and its sampling adjusted counterpart Morisita's index, I_{σ} , are unitless, invariant to sample size, and related analytically to variance in RS, but are sensitive to age-structure (I is even sensitive to mean RS)."

It is surprising that the authors don't mention an earlier paper by Crow and Morton (1955 Evolution 9:202-214), where they define the Index of Variability as $Vk/kbar$, which I refer to here as ϕ . ϕ is arguably the simplest and most direct index of reproductive skew. Using the Poisson approximation to binomial variance, whereby $E(Vk) = kbar$ for random (Wright-Fisher) reproduction, $\phi > 1$ implies

reproductive skew in excess of what is expected under a null model that each adult has an equal opportunity to contribute to the next generation. However, as Crow and Morton point out, if variance in RS is overdispersed, the magnitude of ϕ is very sensitive to k_{bar} . This can be an issue in any experimental design but is particularly problematical if juvenile offspring are sampled from a species with type-III survivorship (e.g., an insect or a fish or a marine invertebrate).

We appreciate the historical background. We chose to build off of the I-index rather than ϕ , as the scaling problems of ϕ are immediately clear. However, in the interest of reflecting the historical development of these ideas, we now cite the indicated Crow and Morton paper, and show how M relates to ϕ in Section 5, lines 380-387.

In that situation, k_{bar} can be \gg the 2 required for a stable population, with the result that ϕ can be very large even if there is only modest reproductive skew in the population. Crow and Morton proposed a simple solution to this problem: ϕ can be rescaled to its expected value at a different mean offspring number using this simple formula:

$$E(\phi_2) = 1 + k_1/k_2 (\phi_1 - 1).$$

where k_1 is mean offspring number in the initial sample, k_2 is the target mean offspring number, ϕ_1 is the initial Index of Variability, and ϕ_2 is the rescaled index. This equation solves for the expected value of ϕ , assuming random mortality of offspring until the target k is reached. Statistically, this is equivalent to randomly sampling fewer offspring in the first place. Setting $k_2 = 2$ yields the expected value of ϕ for a constant population that is exhaustively sampled. There is nothing magical about scaling to $k_2 = 2$, but it is a useful reference point because most populations that persist for any period of time must have a long-term mean offspring number close to 2. Because of the simple relationship between I and ϕ ($I = \phi/k_{bar}$), Crow and Morton's method for rescaling ϕ can also be applied to I to remove the dependence on the sample mean. Curiously, Crow himself did not mention the 1955 variance-rescaling method in either his original 1958 paper or his 1989 Update. Wade and Arnold (1980 Animal Behavior) noted that the Crow and Morton method could be used to scale I to a constant $k = 2$, but subsequent papers of theirs did not follow up on this idea. Recently I have become interested in estimates of age- and sex-specific ϕ_x (e.g, the ratio of variance to mean RS of 6-year old males or 4-year old females). Standard life tables give expected age-specific k_{bar} , but associated age-specific variances or ϕ_x are rarely reported. In analyzing some of the empirical data we compiled, I wondered what people were doing about the dependence of Crow's I on sample size. I found lots of papers noting the problem but no real solutions. It was easy to show that Crow and Morton's equation above removes the dependence of I on sample k_{bar} , but this is not an ideal solution because 1) the result still depends on the choice of k_{bar} to rescale the variance to, and 2) it is not clear whether rescaled I can still be interpreted as the variance of relative fitness, as Crow and Lynch/Walsh have shown applies to raw I.

It also seemed to me important to know whether I computed from raw data is larger than what would be expected due to random sampling variance. It turns out this is easy to do. Since $E(\phi) = 1$ under W-F reproduction and $I = \phi/k_{bar}$, $E(I)$ under W-F is just $1/k_{bar}$. I then constructed a new index which I call $\Delta I = I - E(I_{drift}) = I - 1/k_{bar}$, which removes the dependence of I on sample k without the need for rescaling ϕ . Subsequently, in reviewing the literature I found that this idea had been floated (but abruptly dismissed) by Downhower et al. (1986; cited by the authors). Initially I planned to put all this in an appendix of an empirical paper I was working on. However, eventually I decided to write up a separate paper describing the new ΔI index and pointing out that it could help to directly compare empirical estimates of the Opportunity for Selection from different studies; different samples; samples from different life stages; or males and females when sex ratio is uneven. That manuscript was submitted to a journal in January and a revised version responding to reviewer comments was submitted May 1.

As it would be useful to be able to cite these methodological issues in some other empirical manuscripts I am working on, I also sent the revised version to bioRxiv, where it was posted on May 8 and can be accessed at: <https://biorxiv.org/cgi/content/short/2020.05.06.081224v1>. I note from the authors' response letter that they have done something similar: the PRSLB manuscript was created to move some of the more theoretical/technical details to a separate document that could then be cited in a more empirical study (PNAS paper in revision).

Indeed, this is very similar to our publication experience. We have found Waples' theory paper to be insightful, and well researched. Indeed, the above explanation is exactly why we independently performed the correction from \hat{M} to M in the first equation of our paper. As the reviewer points out—though our derivations are different—we end up correcting the I-index in a similar way *with respect to mean RS dependence*. Waples approach isn't new, however, as the same correction for sampling was done by Morisita (1962) with respect to Crow's I-index, and similar corrections have been described Kokko and Lindstrom (1997), Nonacs (2003), and others for a variety of skew metrics. However, we now cite Waples (2020) on lines 244-251, in place of Kokko and Lindstrom (1997) and Nonacs (2003), as his explanation for the sampling adjustment in the context of the I-index is the most comprehensive we have seen.

At this point, however, our paper and Waples' paper diverge. We develop a further generalization of the I-index that deals with age structure, much like Nonacs' B does, but without introducing sample size dependence. We also develop Bayesian methods to estimate posterior distributions for our metric, rather than relying on point estimates. We introduce an R package with a wide array of functions and documentation, to make the use of M straightforward for the empirical research community. And, we conduct a reanalysis of data from our field as an example to show how the use of problematic skew metrics can muddy scientific inference. The Waples paper, instead, deploys a variety of other robustness checks, and focuses on sampling design and technical details surrounding his field of interest. We think that our papers are very complementary, and that we do not "step on each other toes". We refer readers to Waples' work in several places in our text now, including: lines 149, 244-251, 290-295, and 420-428.

At line 185 the authors define their multinomial index $M(r, t)$ to be the difference between the observed $\hat{M}(r, t)$ and the expected value if RS were distributed as a multinomial outcome with the same \bar{k} and exposure time vector. So, it appears that they and I have independently come up with the same general idea (which actually was proposed but never implemented over 30 years ago by someone else). I don't see this as a problem and don't think it detracts from the usefulness of the present ms, which provides a more general treatment and many other analyses that are apparently of interest to many (but which I don't really feel qualified to comment on). In any case, all of the scaling issues discussed above are in my bioRxiv manuscript, which the authors now have access to.

We appreciate the reviewer's amicability here. We agree that both papers are important contributions to the literature. We agree with the reviewer that we have both independently, and through distinct derivations, arrived at a correction to the I-index that removes dependence on mean RS. We note again, however, that our M index also deals with age/exposure-time dependence, essentially uniting the partial strength of Waples' Δ_I with partial strengths of Nonacs' B.

Waples' paper provides a deeper dive into some scaling issues, biological relationships, and sampling issues related to measuring reproductive skew, that are really only tangential to our goals in this paper. We strongly agree with Waples that sample selection and other methodological choices can impact the usefulness of M and any other metric (see our revised lines 214-264). We now refer readers interested in such topics to his bioRxiv paper. Waples' paper provides an excellent summary of how an index, Δ_I , which is a special case of M , can be deployed.

Additional comments:

The authors are claiming that their method is quite general, but for that to be the case I think there needs to be more discussion about how offspring number is determined (genetic parentage analysis? Behavioral observations?) and when offspring are enumerated, because those can vary dramatically across species and across studies within species. Offspring number is the most direct measure of individual fitness, but juvenile offspring are only a measure of potential fitness because offspring that don't survive to adulthood cannot pass on their parental genes to subsequent generations. If non-random (family correlated) mortality occurs between the time offspring are enumerated and the time they become reproductively active, early life-stage samples will underestimate the realized degree of reproductive skew.

Yes, skew in birth rates is different than skew in recruitment of surviving offspring, and will not reflect differential offspring survival. We are in agreement. We make no assertion that one should measure any particular proxy for fitness. The research question will determine the relevant data-collection and sampling protocols. This issue is largely orthogonal to our argument here, which is, that no matter how careful one's sampling design is, or which fitness proxy one chooses to measure, comparative study of skew using the most commonly used skew metrics, like B, will be confounded by bias introduced by variation in sample/group size. We introduce M to correct such structural dependencies on sample size. We have added lines 214-264, however, outlining the various pitfalls and opportunities associated with different measure of fitness insofar as these are important considerations for making inferences from analyses using the M index.

I also question the generality of the adjustment in the proposed M index to account for heterogeneity in age or 'exposure time' to risk of reproduction. To do this requires knowing or being able to estimate age-specific vital rates (survival and fecundity). The basic derivation of M assumes that fecundity does not vary with age (line 266). The authors also provide a variation that allows for a simple elasticity parameter for age-specific fecundity. These two scenarios might be adequate for the mammalian species the authors are interested in, but they don't come close to capturing the range of patterns of change in fecundity with age that are found in nature (see the figure at the end, reproduced from our 2013 PRSLB paper that compiled life tables from > 60 animal and plant species). What about all of the species with patterns of age-specific fecundity that don't conform to these two assumptions?

Yes, in order to apply M, one needs to measure both a proxy of RS and a proxy of age/'exposure time'. We acknowledge that the most basic formulation of M assumes—implausibly, in the case of humans, for example—that fecundity is proportional to exposure time. We then immediately provide a generalization that relaxes this assumption, and allows for diminishing marginal fecundity to age. This functional form of the age-fecundity relationship is a simple and biologically reasonable assumption for human-type reproduction, but may need to be modified for other species with strikingly different age patterns of reproduction.

In the case of more complicated functional forms linking RS and age, we have now provided a substantially more general solution using a Gaussian Process to formally estimate the shape of the function linking mean RS and age. See section 2 of the supplementary appendix, where we define such a model mathematically, and test its performance using simulated data. This extension is now cited on lines 335-342.

We believe that M is an important generalization of the I-index, and one that will open the door to much more comparative research, especially in the biological, primatological, and anthropological contexts we mention explicitly. It is, of course, possible to point out cases when even M is insufficient to make precise scientific inferences. However, we would argue that M is a substantial

improvement over the measures, like B, currently in widest use. Any of the above critiques could be levied against Nonac's B as well, or any skew measure that seeks to investigate offset from an expected reproductive rate.

From our revised Figure 1, contrasting the performance of 10 of the most widely-used skew metrics, we see that M is the most analytically robust and general metric of its kind. Moreover, our revised version of $M(r,a,b)$, which now uses a Gaussian Process model to estimate the conditional expected value of r_i based on age/exposure time data, is substantially more general than the version of M in the original submission. Our R package now estimates basic M, and M using either elasticity control or Gaussian Process control for the age-specific fertility function. See supplementary appendix, section 2 for details.

Also, the authors go on to say, "In the generalized version of M , if reproductive rates were equal across all individuals within an age-class ..." I interpret this to mean, "if $\phi_x = 1$ for each age and sex" (i.e., if same-age, same-sex individuals act like mini-Wright-Fisher populations with random reproductive success). This might be a realistic assumption for some species but certainly not others. What happens if ϕ_x is overdispersed, as often is the case?

ϕ_x is a real-valued scalar number, not a distribution, so it is not clear what it means to call it "overdispersed". If the reviewer means that the distribution of RS within a given age class is overdispersed (relative to a Poisson), then M will indicate the presence of reproductive skew. In fact, the *quantity of such overdispersion* is precisely what M was created to measure. Specifically, \hat{M} will measure how much residual standardized variance in RS is present after accounting for the variance in RS attributable to variation in age/exposure time. M then measures how much \hat{M} exceeds that which is expected under a multinomial model with the same number of individuals, who have the same exposure time windows, but who have equal expected RS rates per age class.

Another important issue I did not see discussed is what is assumed regarding 1) independence of survival and reproduction, and 2) independence of reproduction across years or seasons? For example, Hill's model for N_e with overlapping generations assumes that reproduction in a given year or season does not affect subsequent survival, nor does it affect expected offspring number in subsequent years. One or both of these assumptions are violated in many species. How, if at all, would violation of these assumptions affect performance of the M index or conclusions of the authors?

Regarding point (1), positive or negative correlations between survival and reproduction will affect M. For example, if fertility and adult mortality were positively correlated due to trade-offs, exposure-time-adjusted M would be higher than M calculated from lifetime RS. This is because the exposure-time-adjusted M would see that high-mortality-high-fertility phenotypes reproduce at a very high rate while alive compared to lower-mortality-lower-fertility phenotypes. If fertility and adult mortality were negatively correlated due to phenotypic correlation, exposure-time-adjusted M would be lower than M calculated from lifetime RS. This is because the low-fertility phenotypes that die earlier will have shorter exposure times, so they wouldn't look quite so pitiful next to the long-lived-high-fertility individuals.

This tells us again that examining skew from different perspectives, or with different data sub-sets, can be instructive. Skew in reproductive rates while alive and potentially reproductive is relevant to some questions. Skew in lifetime RS is relevant to others, e.g., the combined effect of differential fertility and adult mortality. Skew in life-time recruitment (the number of offspring that grow to adulthood) further conveys information about the combined effect of fertility, adult mortality, and offspring mortality. These are all interesting and valid things to measure and study. We make no

argument for which should be done, as this depends on the question. See our new text on lines 252-263.

We note that the questions raised by the reviewer are not questions about the validity of M . Instead, they address which data should be fed into M . For example, should we use offspring ever born as our proxy of RS? Or offspring recruited to reproductive age as our proxy of RS? Should we include only individuals with completed RS outcomes (i.e. those with complete birth-to-death records)? Should we control for exposure time (e.g., which would reflect variation in survival, assuming only those with complete life-time records are used)? Our response, is that these choices will all depend on the research question at hand. We note, however, that M , unlike previous metrics like the I index, Morisita's index, or the B index, can deal with each of these cases. For example, we could use data from those with complete life-time records. We could, for example, calculate M from reproductively complete individuals, with no adjustment for exposure time, and then calculate M with adjustment for exposure time, in order to unpack the role of differential survival on skew. It is possible that some systems would exhibit little skew in RS while living, but substantial skew in lifetime RS, if survival is highly variable. M is a tool to study such possibilities; it does not *limit* us to only studying skew in RS rate while alive. Our R package provides functionality for each kind of analysis, simply by changing which data are fed into M and whether exposure time data is included, or fixed to a constant value.

Regarding point (2), we make no assumptions with regard to such underlying mechanisms. M simply measures how much the standardized variance in RS differs from what would be expected under a multinomial model where reproduction has an expectation that is proportional to exposure time. Models aiming to estimate other quantities (like N_e) from standardized measures of variation in RS might be sensitive to the details of the generative model in the way the reviewer describes. We don't use M to try and estimate N_e , so this issue is unrelated to our argument. We see that the reviewer has a section on this issue in his bioRxiv paper, as estimating such values is a goal of his work. We defer interested parties to his work on the topic in line 149.

I am not clear what exactly one is left with after data on lifetime RS are adjusted to remove the heterogeneity in longevity. Is the result some sort of average value of I that applies to just one year/season/reproductive interval?

What is left is heterogeneity in reproductive rates while alive, which is the specific focus of our empirical paper. Other research questions that are not tied to measurement of inequality in reproductive rate while alive may be best answered with other tools, or by applying M to different sub-samples of data (i.e., those with completed reproductive histories). If inequality in reproductive rate is not the focus of a research question, M can still be deployed, but with exposure time held fixed. This would reduce to M to Δ_i . Different research questions can be addressed by different sample selection methods and data inputs. We are agnostic with respect to what researchers should do; this all depends on the research question. See lines 214-264.

Also, do results depend on whether reproduction is assumed to be continuous (as in humans) or corresponds to Caswell's seasonal birth-pulse model?

As before, M is simply a metric that measures how much the standardized variance in RS differs from what would be expected under a multinomial model where reproduction has an expectation that is proportional to effective exposure time. Our derivation makes no assumption about seasonal, or other, timing. Further generalizations of M , dealing with more complicated models for the second term in our Eq. 3.1, or the drift term in Waples' paper, might be useful to study such dynamics, but significantly exceed the scope of the current paper.

I hoped to try out the SkewCalc software and was glad to see from the Appendix that just 2 lines of code are needed for the installation. However, the installation requires that Rtools be installed, which turns out to be non-trivial and can't be accomplished with simply an "install.packages" command. I followed the instructions on the Rtools website and it seemed to install correctly, but I get a series of error messages when I then try to install SkewCalc. I didn't have the time or energy to try to work through the problems so I gave up.

We thank the reviewer for taking the time to try and check out our software. Our statistical workflow relies on what is now the gold-standard software for Markov Chain Monte Carlo, Stan. Stan requires a C++ tool chain, which RTools provides. Once RTools is installed properly, and RStan and devtools packages loaded into R, then SkewCalc runs fine (and we have tested on several machines). We recommend following the setup instructions for RStan here: <https://github.com/stan-dev/rstan/wiki/RStan-Getting-Started>

Once RStan is installed correctly, we foresee no issues with SkewCalc installation. But, unfortunately, without installing Stan, one can't run Stan models. Our software builds custom C++ models, with almost no user input, but they can't be run if the user doesn't have the prerequisite programs needed to compile such models.

Minor comment:

"Reproductive skew is common across many animal societies." True, but the same could be said about plants, so why limit the comment to animals? As noted above, the full range of diversity in reproductive skew does not appear to be accommodated by the M index, despite the authors' claims for its generality.

We don't claim M to be so general that it can be used across any conceivable context. Rather, we show that there are several desiderata in a comparable skew metric, especially with respect to the animal societies that are the focus of our own empirical research and most published work on reproductive skew: 1) sample size invariance, 2) mean RS invariance, and 3) age-structure invariance. Just as we argued to Nonacs about B—that it does not permit comparative study of reproductive skew due to bias introduced by sample size variance across sites, even though it controls for age structure (i.e., it meets desideratum 3 but not desideratum 1)—we argue to Waples that Δ_i (just like its original formulation as Morisita's I_o , which we had previously cited) meets desideratum 1 but not desideratum 3. We prove that M meets all three desiderata. Maybe M is not fully general across all conceivable systems, but neither is Δ_i , which is a special case of M.

As we show in Fig. 1, which has been updated to include Q and Δ_i , M is still the most robust metric we have seen for comparison of skew across groups in the domains of research we focus on—mammals, primates, and humans. We therefore believe that M constitutes an important contribution to the literature.

Note: I see from the response letter that the authors have compiled RS data for a large number of mammals. Are those data publicly available? If so it would be very useful to be able to access them. They relate directly to this request a colleague and I distributed widely last summer, through list serves and email.

We have not made these data sets publically available yet, but we will do so as soon as possible.

Appendix C

Associate Editor. Comments to Author:

I think the authors made a good job in further improving the paper. I only have a minor thing. I 201—check formatting

Fixed.

Reviewer(s). Comments to Author. Referee: 4

The authors provided a detailed response to my comments on the previous version, and the revised manuscript is much improved. In particular, since I deal with similar issues from a population genetics perspective, it is easier to understand the anthropological perspective in the revised version; the previous version was skimpy on those details and basically assumed that the anthropological perspective is the only way to think about these issues. That should make the manuscript more accessible to the general reader. If I wanted to quibble, I might point out that although the existence of other perspectives is acknowledged in the revised version, the population genetics perspective is not really developed in any meaningful way.

We are glad to hear that we have achieved a more ecumenical presentation of our work.

My bioRxiv paper on the delta I index is now published in Evolution (available at <http://dx.doi.org/10.1111/evo.14061>), so that citation can be updated. All the major results and conclusions remain unchanged, but the published version contains a couple of additions that the authors might look at to see whether there is anything useful relative to their paper on the multinomial index.

We have updated the citation.

1) In response to reviewer/editor comments about notation, I redid the analytical model in terms of a generalized Wright-Fisher model of reproduction. In the standard WF model, each parent contributes equally to a large (~infinite) pool of gametes that unite at random to form the offspring generation. This can be explicitly modeled in a computer by allowing each offspring to 'choose' its parents randomly and with replacement from the pool of adults, all of whom have an equal probability of being selected at each draw. The generalized WF model differs in allowing unequal contributions to the initial gamete pool, which can be modeled by allowing different adults to have different probabilities of being a parent. These probabilities can be expressed as a vector of weights, W . The potential relevance of this is that the expected value of Crow's I and delta I can be expressed as simple functions of the squared CV of W .

This appears directed at the editor about Waples' work. We make no changes to our paper here.

2) Like the authors, the major focus in my paper was on finding a way to adjust for effects of mean offspring number in a sample, which can reflect experimental design, sampling effort, logistical constraints, and other factors that represent noise. However, a colleague of mine who works with fantastic long-term datasets on reproductive success of great tits provided a novel perspective: in their studies, they basically inventory the entire population every year, but sample size of recruits varies across years as a result of demographic and environmental fluctuations. In this situation, temporal variation in mean offspring number reflects temporal changes in mean fitness in the population as a whole, not just in the sample. In this special case, therefore, removing this effect using the delta I index could be obscuring a true biological signal related to intensity of selection. I added a short para near the end of Discussion pointing out this issue, and the authors might consider whether this issue is also potentially relevant to the M index.

Yes indeed. This is an issue we had raised in our paper before when we stated: “The absolute measure of skew might be more evolutionary relevant for some problems, however, so the use of $\dot{M}(r,t)$ or $M(r,t)$ needs to be based on context.” We ended up cutting this statement during our last revision, and wrapping the idea up into another more general paragraph.

We agree with the reviewer here, however, so we have added this statement back in, citing the specific example above: “The absolute measure of skew might be more relevant for some problems—for example, if change in reproductive skew is measured using full census data from a single population over time, see Waples (2020)—so use of $\dot{M}(r,t)$ or $M(r,t)$ needs to be based on context.”

L 61: “One desiderata” -> one desideratum

Good catch. Fixed.